# OmniSTVG: Toward Spatio-Temporal Omni-Object Video Grounding

**Jiali Yao**[1*]   **Xin Gu**[2*]   **Xinran Deng**[2]   **Mengrui Dai**[3]   **Bing Fan**[4]
**Zhipeng Zhang**[5]   **Yan Huang**[4]   **Heng Fan**[4†]   **Libo Zhang**[6†]

[1]Hangzhou Institute for Advanced Study, University of Chinese Academy of Sciences
[2]University of Chinese Academy of Sciences   [3]North China University of Technology
[4]University of North Texas   [5]Shanghai Jiao Tong University
[6]Institute of Software Chinese Academy of Sciences
yaojiali24@mails.ucas.ac.cn, libo@iscas.ac.cn

## ABSTRACT

We introduce spatio-temporal omni-object video grounding, dubbed **OmniSTVG**, a new STVG task aiming to localize spatially and temporally *all* targets mentioned in the textual query within videos. Compared to classic STVG locating only a single target, OmniSTVG enables localization of not only an arbitrary number of text-referred targets but also their interacting counterparts in the query from the video, making it more flexible and practical in real scenarios for comprehensive understanding. In order to facilitate exploration of OmniSTVG, we propose **BOSTVG**, a large-scale benchmark dedicated to OmniSTVG. Specifically, BOSTVG contains 10,018 videos with 10.2M frames and covers a wide selection of 287 classes from diverse scenarios. Each sequence, paired with a free-form textual query, encompasses a varying number of targets ranging from 1 to 10. To ensure high quality, each video is manually annotated with meticulous inspection and refinement. To our best knowledge, BOSTVG, to date, is the first and the largest benchmark for OmniSTVG. To encourage future research, we present a simple yet effective approach, named **OmniTube**, which, drawing inspiration from Transformer-based STVG methods, is specially designed for OmniSTVG and demonstrates promising results. By releasing BOSTVG, we hope to go beyond classic STVG by locating every object appearing in the query for more comprehensive understanding, opening up a new direction for STVG. Our project is released at https://jellyyao3000.github.io/OmniSTVG/.

## 1 INTRODUCTION

*Query:* *A man* *in black catches the soccer*

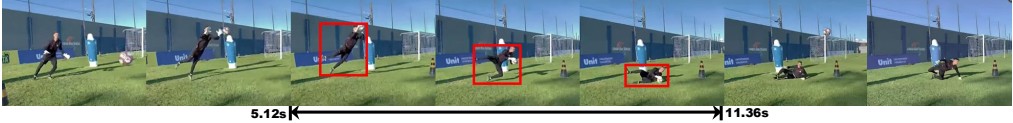

(a) Current spatio-temporal video grounding (**STVG**) localizing *a single target*

*Query:* *As* *the elephant* *steps on* *seesaw*, *the man* *holding* *a basketball* *jumps up*

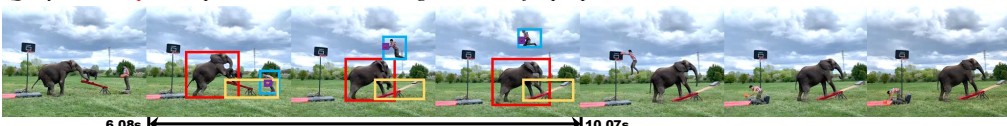

(b) The proposed spatio-temporal omni-object video grounding (**OmniSTVG**) localizing *all mentioned targets*

Figure 1: Comparison of existing *STVG* that localizes a single object in the query (a) and our *OmniSTVG* localizing all objects in the query (b). The object in the query and its corresponding spatio-temporal tube in the video is highlighted using the same color. *Best viewed in color for all figures.*

---

*Equal contribution   †Equal advising and corresponding authors

Spatio-temporal video grounding (STVG) (Zhang et al., 2020b) has been one of the crucial problems in multimodal video understanding. Given a free-form textual query, it aims at locating the target of interest in space and time from the video (see Fig. 1 (a)). Owing to its crucial applications, including human-machine interaction, robotics, *etc.*, STVG has gained increasing interest in recent years (Su et al., 2021; Yang et al., 2022; Jin et al., 2022; Lin et al., 2023; Wasim et al., 2024; Gu et al., 2025).

While significant progress has been witnessed, localizing only a single object, as it is done in current STVG, is *insufficient* for video understanding in many real-world scenarios. For instance, in daily life, since an event or an activity often involves various objects (see Fig. 1 (b)), it is common a textual query contains *multiple* targets of interest (*e.g.*, "*elephant* and *man*" in query in Fig. 1 (b)). For such query, it is essential to localize *every* queried objects, spatially and temporally, within the video. Yet, existing STVG localizes only a single target in query (see Fig. 1 (a)), and hence is restricted in multi-objective query localization, degrading practicability. To alleviate this, a straightforward solution is to employ the STVG model repeatedly for multiple single-object textual queries. Nevertheless, this markedly increases the computational burden, leading to the lack of the scalability in existing STVG.

In addition to the restriction in locating multiple queried targets, another limitation of current STVG is the *ignorance* of interacting counterparts for queried objects. In practice, the object of interest is usually *not alone* but interacts with other targets (see textual queries in Fig. 1 (a) and (b)). Localization of objects of interest, together with the interacting counterparts, provides richer contextual information for the objects, and thus allows more comprehensive spatio-temporal understanding of the video, which greatly benefits many applications, including surveillance, robotics, sport analysis, and so forth. In existing STVG, however, the crucial interacting counterparts are often neglected in localization, restricting the classic STVG for more comprehensive analysis.

**Contribution.** To mitigate aforementioned limitations of current STVG, the key is to have the ability to locate *every* target mentioned in the textual query, much like how humans do. Thus motivated, we introduce a new type of STVG task, dubbed *Spatio-Temporal Omni-Object Video-Grounding* (or *OmniSTVG*). Different from existing STVG locating only a *single* object, OmniSTVG aims at localizing *all* targets mentioned in the given query within the video. For each object in the query, a spatio-temporal tube is predicted as the localization result. By doing so, OmniSTVG enables localization of *not only* arbitrary number (*e.g.*, one or multiple) of targets of interest *but also* their interacting counterparts in the video, which simultaneously resolves two limitations of current STVG and therefore leads to more practical applications. It is worth noting, OmniSTVG is a natural extension of classic STVG task, aiming to further push its frontier for more comprehensive multimodal video understanding. Concept-wise, OmniSTVG, to some extent, is inspired by the idea of *segmentation anything* (SA) (Kirillov et al., 2023). The difference is, SA aims to segment any regions in an image, while OmniSTVG locates any mentioned objects in the textual query from an untrimmed video.

To facilitate exploration of OmniSTVG, we propose *BOSTVG*, a novel large-scale dataset dedicated to spatio-temporal omni-object video grounding. Specifically, our BOSTVG contains 10,018 videos with 10.2 million frames, and covers a wide selection of 287 categories from diverse scenarios. Each sequence in BOSTVG, paired with a free-form textual query, contains a varying number of objects to localize, ranging from 1 to 10 with an average of 2.4. Each object is manually annotated with a spatio-temporal tube (*i.e.*, a set of bounding boxes). To ensure high quality, all the tube annotations in each sequence are carefully inspected and refined when needed through multiple rounds. To our best knowledge, BOSTVG by far is the *first* and the *largest* benchmark dedicated to OmniSTVG.

Furthermore, to encourage future research in developing OmniSTVG methods on BOSTVG, we propose a simple yet effective model, dubbed *OmniTube*. Specifically, OmniTube is built upon current Transformer-based STVG method (Yang et al., 2022). It comprises a multimodal encoder for video and text feature fusion and a decoder for localization. Different from current STVG methods (Yang et al., 2022; Gu et al., 2024; Jin et al., 2022; Lin et al., 2023) that locate only a single target, our OmniTube learns *simultaneously* multiple sets of object queries in the decoder to ground *all* objects in the video. To improve localization, we apply visual information in video guided by textual feature to generate queries, which benefits learning better query features for target localization. To form a spatio-temporal box tube for each target, a simple strategy is introduced to match detection results across different frames from the video. Despite simplicity, OmniTube shows promising results and expects to provide a reference for future research on our OmniSTVG task.

Table 1: Comparison of BOSTVG with other benchmarks. SO: Single-Object; MO: Multi-Object; AO: All-Object. ¶: Since DVD-ST is not released, we report the statistics from its original paper.

| Benchmark | Videos | Object classes | Mean frames | Total frames | Total dur. | Min obj. | Mean obj. | Max obj. | Total obj. | Num. of queries | Dataset focus |
|---|---|---|---|---|---|---|---|---|---|---|---|
| STPR (Yamaguchi et al., 2017) | 5,293 | 1 | 260 | 1.4M | 14 hrs | 1 | 1.0 | 1 | 5,828 | 30,365 | SO |
| VID-Sentence (Chen et al., 2019) | 5,318 | 30 | 294 | 2.3M | 21 hrs | 1 | 1.0 | 1 | 7,654 | 7,654 | SO |
| VidSTG (Zhang et al., 2020b) | 6,924 | 79 | 798 | 5.5M | 53 hrs | 1 | 1.0 | 1 | 6,924 | 99,943 | SO |
| HCSTVG-v1 (Tang et al., 2021) | 5,660 | 1 | 522 | 3.0M | 31 hrs | 1 | 1.0 | 1 | 5,660 | 5,660 | SO |
| HCSTVG-v2 (Tang et al., 2021) | 16,544 | 1 | 522 | 8.6M | 92 hrs | 1 | 1.0 | 1 | 16,544 | 16,544 | SO |
| DVD-ST (Ji et al., 2024)¶ | 2,750 | 163 | - | - | - | 0 | 1.8 | 12 | 4,950 | 5,734 | SO, MO |
| **BOSTVG** (ours) | 10,018 | 287 | 1,014 | 10.2M | 102 hrs | 1 | 2.4 | 10 | 24,175 | 10,018 | AO |

We notice a concurrent work (Ji et al., 2024) which has similar spirit with this work by supporting grounding multiples in videos. Compared to (Ji et al., 2024), our work mainly *differs* in three aspects. First, *concept-wise*, OmniSTVG grounds *all* objects mentioned in query, while the work of (Ji et al., 2024) localizes only *partial* targets, leading to limitations in comprehensive understanding. Second, *task-* and *method-wise*, the work of (Ji et al., 2024) locates only target objects of the *same* class, while our OmniSTVG and OmniTube enable the localization of objects of *different* categories, making it more flexible as well as practical. Third, *dataset-wise*, our BOSTVG contains 10,018 videos, which is much larger than the dataset in (Ji et al., 2024) with 2,750 sequences.

In summary, we make the following contributions: ♠ We propose OmniSTVG, a new STVG task that locates all objects mentioned in the query toward more flexible and comprehensive understanding; ♥ We present BOSTVG, a large-scale dataset with 10,018 videos and more than 10 million frames from 287 categories for OmniSTVG; ♣ We introduce OmniTube, a simple but effective method to facilitate more future research of OmniSTVG; ♦ We demonstrate that OmniTube achieves promising performance, aiming to offer a reference and provide guidance for future research on OmniSTVG.

## 2 RELATED WORK

**STVG Benchmarks.** Benchmarks play an important role for the development of STVG. The work of (Yamaguchi et al., 2017) introduces the STPR for spatially and temporally grounding pedestrians from trimmed videos. Similarly, VID-Sentence (Chen et al., 2019) is proposed for grounding within trimmed videos, but comprises more classes. For more practical STVG, HCSTVG-v1 (Tang et al., 2021) is presented to locate human objects in untrimmed videos, making it more challenging. Later, HCSTVG-v2 (Tang et al., 2021) is introduced via expanding from HCSTVG-v1 using extra videos. Different from HCSTVG-v1/v2, VidSTG (Zhang et al., 2020b), collected from VidOR (Shang et al., 2019) for video object relation detection, aims at spatio-temporal video grounding from both declarative and interrogative sentences. In addition, besides human category, VidSTG offers other classes in the query and video for localization, aiming at generic STVG. Unlike previous datasets only for single-target localization, the recent DVD-ST (Ji et al., 2024) introduces a platform for grounding multiple targets while ignoring interacting counterparts in localization. ***Different*** from all the aforementioned benchmarks, BOSTVG is specially developed for a new STVG task, OmniSTVG, which aims to ground *all* targets mentioned in the textual query. Therefore, in our BOSTVG, each target in the query is annotated with a spatio-temporal tube (see Fig. 1 (b) again for an example in BOSTVG).

**STVG Algorithms.** STVG algorithms have witnessed great progress recently. Early methods (Tang et al., 2021; Zhang et al., 2020a;b) typically adopt a two-stage pipeline, which first detects candidate region proposals with a pre-trained detector (Ren et al., 2015) and then finds correct region proposals with an extra model. Despite straightforwardness, these two-stage methods heavily rely on the pre-trained detection model, and their performance is thus restricted by the capacity of the used detector. To overcome this limitation, recent STVG methods (Su et al., 2021; Gu et al., 2024; Yang et al., 2022; Jin et al., 2022; Lin et al., 2023; Wasim et al., 2024; Gu et al., 2025), inspired by DETR (Carion et al., 2020), switch to a one-stage design directly predicting a spatial-temporal tube for localization using Transformer (Vaswani et al., 2017), without adopting any external detectors. Compared to two-stage methods, such one-stage framework shows superior performance for its end-to-end training pipeline. Our OmniTube is also a one-stage Transformer-based architecture. Nonetheless, ***different*** from the aforementioned approaches that localize only a *single* target from the video, our OmniTube aims to locate *all* objects mentioned in the query for more comprehensive multimodal video understanding.

## 3  THE PROPOSED BOSTVG

### 3.1  DESIGN PRINCIPLE

BOSTVG aims to offer a dedicated platform to facilitate the development of OmniSTVG. For such a purpose, we follow several principles in developing BOSTVG: **(i)** ***Dedicated benchmark.*** The motivation of BOSTVG is to provide a novel benchmark dedicated to OmniSTVG. The video and its paired textual query are required to comprise a varying number of targets (*e.g.*, one or multiple) to localize, aligning with the goal of OmniSTVG. **(ii)** ***Large scale.*** Developing the deep learning-based methods for OmniSTVG requires abundant training samples. Besides training, a real system needs evaluation on various situations. Hence we expect BOSTVG to contain at least 10K videos with each corresponding to a textual query, which benefits both large-scale training and assessment of deep models. **(iii)** ***Diverse object classes.*** An important aim of BOSTVG is to facilitate the development of general OmniSTVG models that can locate target objects from different classes. To this end, the new dataset expects to contain at least 200 categories, collected from various scenarios, for grounding. **(iv)** ***High quality.*** High-quality annotations are essential for a benchmark in both training and evaluation. To ensure the high quality, each video of BOSTVG is manually labeled with precise spatial-temporal box tubes through multiple rounds of inspections and refinements.

### 3.2  DATA ACQUISITION

BOSTVG aims to foster general and comprehensive spatial-temporal video grounding by containing rich classes from diverse scenarios. To this end, 287 categories that are appropriate for OmniSTVG are selected in BOSTVG. These object classes are chosen from different sources, mainly including ImageNet (Deng et al., 2009) and V3Det (Wang et al., 2023), and organized in a coarse-to-fine hierarchical structure. Due to limited space, we show detailed classes in the ***supplementary material***.

After determining all object categories in our BOSTVG, we then search for raw videos of each class under various scenarios from YouTube, currently the largest and most popular video platform with many real-world videos. All videos are collected under the *Creative Commons License* and used for research purpose only. Initially, we have collected over 15K sequences using keywords aligned with object classes. Then, we conduct careful inspections on each video sequence to verify its suitability for OmniSTVG. More specifically, if there is at least one video clip suitable for our task, we keep this video; otherwise, we discard the video. This process is carried out by our experts (*e.g.*, students working on related field). For the qualified videos, we select one clip from each of them. Eventually, we gather 10,018 videos for BOSTVG, with each provided a textual query by our experts.

Finally, we create a large-scale benchmark, called BOSTVG, for OmniSTVG. BOSTVG covers 287 classes and contains 10,018 videos with 10.2 million frames from diverse scenarios. Its average video length is 1,014 frames. Each video contains a varying number of targets ranging from 1 to 10. It is worth noting, we do not consider the case of *none* target in the video in BOSTVG and focus on localizing targets that appear within the video. Tab. 1 summarizes BOSTVG and its comparison to other benchmarks. Due to limited space, we provide detailed discussions of maintenance, ethical issue, and responsible usage of BOSTVG in the ***supplementary material***.

### 3.3  DATA ANNOTATION

In BOSTVG, our experts first generate a textual query for each video. After that, we offer two types of annotations based on the query, including start and end timestamps for temporal localization and bounding box tubes of objects for spatial localization. Specifically, given a pair of video and textual query, we first identify a temporal segment in the video that corresponds to the query description, and mark it with start and end timestamps. Afterwards, we label each object mentioned in query with a consistent spatio-temporal tube formed by a set of boxes in each frame of temporal segment.

To ensure high-quality annotations of BOSTVG, we use a multi-round strategy. More specifically, a few experts who work on related problems first manually annotate the start and end timestamps for each sequence. Then, the temporal segment of each video, marked by start and end timestamps, is manually labeled by an annotation team that is formed by a few volunteers and an expert. After this initial round, the spatio-temporal tube annotations will be sent to a validation team formed by three experts for inspection. If the initial annotations are not unanimously agreed by all experts, they will

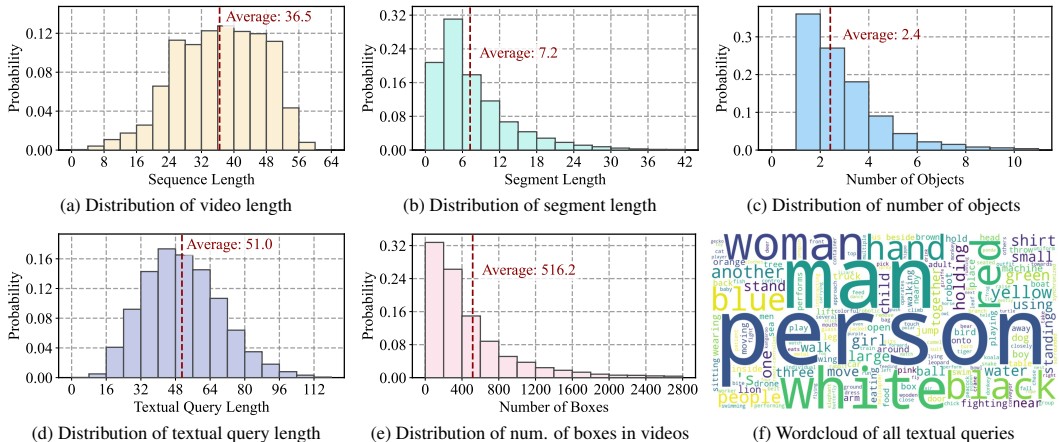

Figure 2: Representative statistics on BOSTVG, including distributions of video length (in seconds) in (a), temporal segment length (in seconds) in (b), number of target objects for grounding in (c), query length (in characters) in (d), number of boxes in videos in (e), and wordcloud of queries in (f).

be returned back to the original labeling team for refinement. We repeat this inspection-refinement process until the annotations of all videos are qualified. Due to limited space, we show the pipeline of our annotation process in the ***supplementary material***. Fig. 1 (b) displays an annotation example in BOSTVG, and more can be seen in the ***supplementary material*** due to limited space.

**Annotation accuracy analysis.** To analyze our annotation accuracy, we randomly select 100 videos from BOSTVG and ask an independent group of external experts to inspect and re-label them. Then, we compute the Intersection over Union (IoU) of new and original spatio-temporal tubes. The IoU for these selected sequences is 0.90, which validates accuracy and reliability of our annotations.

**Statistics of annotation.** To better understand BOSTVG, we display some statics in Fig. 2, including distributions of video length, temporal segment length, and number of objects, textual query length, number of boxes in videos, and the query wordcloud. From Fig. 2 (c), we can see that most videos contain 1 to 6 objects, which is close to real scenarios and makes BOSTVG more suitable for practical applications. From Fig. 2 (e), we can observe that the average number of boxes in each video is 516.2, showing the challenge of BOSTVG in localizing more objects.

## 3.4 DATASET SPLIT AND EVALUATION METRIC

**Dataset Split.** BOSTVG contains 10,018 videos in total. Among them, 8,106 videos are selected for the training set, dubbed $BOSTVG_{Tra}$, and the rest 1,912 for test set, dubbed $BOSTVG_{Tst}$. Tab. 2 displays the comparison between training and test sets. Please note, in split, we try our best to keep

Table 2: Comparison of training and test sets.

| | Videos | Mean frames | Total frames | Mean obj. | Total obj. |
|---|---|---|---|---|---|
| $BOSTVG_{Tra}$ | 8,106 | 1,014 | 8.22M | 2.4 | 19,567 |
| $BOSTVG_{Tst}$ | 1,912 | 1,015 | 1.94M | 2.4 | 4,608 |

distributions of training and test sets similar. For both training and test sets, each video is paired with a textual involving a varying number of targets for grounding, meeting the aim of OmniSTVG.

In addition, to enable in-depth analysis, we further divide the test set into three subsets based on the number of targets in the video, including $BOSTVG_{Tst}$-Low with 1-3 targets in each video (1,566 samples), $BOSTVG_{Tst}$-Medium with 4-6 targets in each video (273 samples), and $BOSTVG_{Tst}$-High with more than 7 targets in each video (73 samples).

**Evaluation Metric.** Following current benchmarks (Zhang et al., 2020b), we utilize multiple metrics, including $m\_tIoU$, $m\_vIoU$, and $vIoU@R$ for evaluation. Please note, the calculation of m_vIoU and vIoU@R here needs to consider all spatial-temporal box tubes in the video, instead of a single one as in other benchmarks, because the video in our BOSTVG may contain multiple objects. Due to limited space, please refer to the ***supplementary material*** for the detailed formulations of metrics.

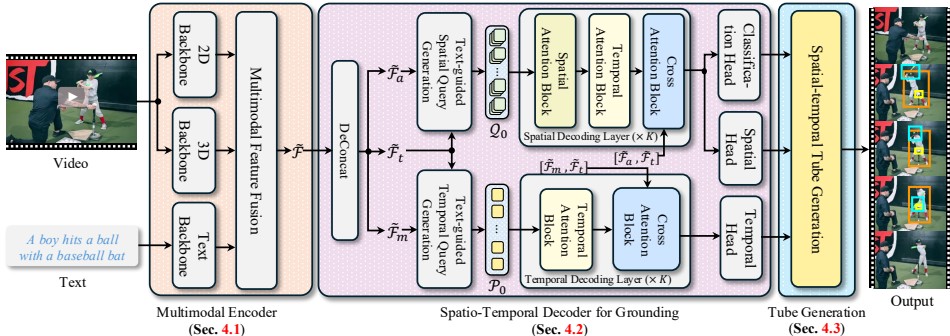

Figure 3: Overview of OmniTube, which contains a multimodal encoder, a spatio-temporal decoder, and a spatial-temporal tube generation module to locate all mentioned targets in the textual query.

# 4 OMNITUBE: A NEW METHOD FOR OMNISTVG

**Overview.** We propose OmniTube, a new framework specially designed for OmniSTVG. Similar to current STVG models (Yang et al., 2022; Jin et al., 2022; Lin et al., 2023), inspired by DETR (Carion et al., 2020), OmniTube adopts an encoder-decoder architecture. As shown in Fig. 3, OmniTube mainly consists of three components, including a multimodal encoder for feature extraction and fusion (Sec. 4.1), a spatio-temporal decoder for target object position learning (Sec. 4.2), and a box tube generation module (Sec. 4.3), to localize all mentioned objects in the text for OmniSTVG.

## 4.1 MULTIMODAL ENCODER

Given the video and the text, the multimodal encoder first extracts their features and then performs multimodal feature fusion as described in the following.

**Feature Extraction.** Given the video, we sample $N_v$ frames from it and then adopt ResNet-101 (He et al., 2016) and VidSwin (Liu et al., 2022a) for 2D appearance and 3D motion feature extraction. The appearance feature is represented as $\mathcal{F}_a = \{f_i^a\}_{i=1}^{N_v}$, where $f_i^a \in \mathbb{R}^{H \times W \times D_a}$ with $H$, $W$ and $D_a$ the height, width, and channel dimensions, and the motion feature as $\mathcal{F}_m = \{f_i^m\}_{i=1}^{N_v}$, where $f_i^m \in \mathbb{R}^{H \times W \times D_m}$ with $D_m$ the channel dimension. For the text, we use RoBERTa (Liu et al., 2019) for feature extraction. We tokenize it to a sequence with $N_t$ words, and then apply RoBERTa on the sequence to produce textual feature $\mathcal{F}_t = \{f_i^t\}_{i=1}^{N_t}$, where $f_i^t \in \mathbb{R}^{D_t}$ with $D_t$ the feature channel.

**Multimodal Feature Fusion.** We generate the multimodal feature by fusing appearance, motion, and textual features. Similar to (Gu et al., 2024), we first project them to the same channel dimension $D$ and then concatenate them to obtain the initial multimodal feature $\mathcal{F} = \{f_i\}_{i=1}^{N_v}$ where $f_i = \texttt{Concat}(\texttt{Ln}(f_i^a), \texttt{Ln}(f_i^m), \texttt{Ln}(f_i^t))$ is the multimodal feature in frame $i$. Then, we fuse the features using the self-attention encoder (Waswani et al., 2017) to generate the multimodal feature $\tilde{\mathcal{F}} = \texttt{SAEncoder}(\mathcal{F} + \mathcal{E}_{\text{pos}} + \mathcal{E}_{\text{type}})$, where $\mathcal{E}_{\text{pos}}$ and $\mathcal{E}_{\text{type}}$ denote position and type embeddings and $\texttt{SAEncoder}(\cdot)$ is self-attention encoder with $L$ ($L$=6) standard self-attention encoder blocks. Due to space limitation, please see its architecture in the ***supplementary material***.

## 4.2 SPATIO-TEMPORAL DECODER FOR GROUNDING

To obtain target position information from multimodal feature $\tilde{\mathcal{F}}$, we design a spatio-temporal decoder composed of a spatial decoder and a temporal decoder. The former learns spatial information for all objects in the text, while the later aims to obtain the temporal information for grounding.

### 4.2.1 SPATIAL OMNI-OBJECT DECODER FOR GROUNDING

**Text-guided Spatial Query Generation.** Unlike current STVG methods locating a single target, OmniTube localizes all objects in text, and thus introduces multiple queries for each frame. To explore target cue for better localization, we leverage text as guidance to select target-relevant features in video for generating object queries. To this end, we first extract features from $\tilde{\mathcal{F}}$ by deconcatena-

tion $[\tilde{\mathcal{F}}_a, \tilde{\mathcal{F}}_m, \tilde{\mathcal{F}}_t]$=DeConcat($\tilde{\mathcal{F}}$), where $\tilde{\mathcal{F}}_a/\tilde{\mathcal{F}}_m/\tilde{\mathcal{F}}_t$ are appearance/motion/textual features. Then, we utilize appearance and textual features to generate the spatial queries. Specifically, we first average textual feature $\tilde{\mathcal{F}}_t$ via $\bar{\mathcal{F}}_t$=Avg($\tilde{\mathcal{F}}_t$). Then, we calculate similarity between $\bar{\mathcal{F}}_t$ and $\tilde{\mathcal{F}}_a$, and adopt the $M$ most similar features to generate initial query $\mathcal{Q}_0$ by average pooling, as follows,

$$\mathcal{Q}_0 = \{\{q_{i,j}^0\}_{j=1}^{N_q}\}_{i=1}^{N_v}, \quad q_{i,j}^0 = \texttt{AvgPooling}(\texttt{Top}_\texttt{M}(\texttt{Sim}(\tilde{\mathcal{F}}_a, \bar{\mathcal{F}}_t)))$$

where $q_{i,j}^0$ denotes the feature of the $j^{\text{th}}$ query in frame $i$, and $N_q$ is the number of queries per frame. $\texttt{Sim}()$ and $\texttt{Top}_\texttt{M}()$ are the operations to calculate the similarities and to pick up top $M$ elements, respectively. Compared with previous approaches, our text-guided object queries by exploring target-specific cues can effectively improve localization as shown in our experiments.

**Spatial Decoding.** After generating $\mathcal{Q}_0$, we send it to spatial decoder with $K$ ($K$=6) layers for interaction with the multimodal feature. To enhance queries, in each layer we design two spatial and temporal attention blocks to capture their spatial and temporal relation before interacting with the multimodal feature. Specifically, let $\mathcal{Q}_{k-1}$ represent query features sent to the $k^{\text{th}}$ ($1 \leq k \leq K$) layer for learning $\mathcal{Q}_k$. We first perform spatial attention on queries of the same frame, as follows,

$$\{\hat{q}_{i,j}^{k-1}\}_{j=1}^{N_q} = \texttt{SABlock}(\{q_{i,j}^{k-1}\}_{j=1}^{N_q}) \quad i = 1, 2, \cdots, N_v$$

where $\hat{\mathcal{Q}}_{k-1}$=$\{\{\hat{q}_{i,j}^{k-1}\}_{j=1}^{N_q}\}_{i=1}^{N_v}$ is query feature after spatial attention. $\texttt{SABlock}(\cdot)$ is spatial attention block implemented by self-attention as in ***supplementary material***. After this, to capture temporal relation, we use temporal attention on query features of the same object across different frames via

$$\{\tilde{q}_{i,j}^{k-1}\}_{j=1}^{N_q} = \texttt{TABlock}(\{\hat{q}_{i,j}^{k-1}\}_{i=1}^{N_v}) \quad j = 1, 2, \cdots, N_q$$

where $\tilde{\mathcal{Q}}_{k-1}$=$\{\{\tilde{q}_{i,j}^{k-1}\}_{j=1}^{N_q}\}_{i=1}^{N_v}$ is the query features after the temporal attention block $\texttt{TABlock}(\cdot)$ implemented with self-attention as in the ***supplementary material***.

Next, we learn the spatial position of objects by interacting queries with multimodal feature. In OmniTube, spatial localization leverages the appearance and text features. Specifically, we interact $\tilde{\mathcal{Q}}_{k-1}$ with the multimodal feature via cross-attention for learning $\tilde{\mathcal{Q}}_k$, as follows,

$$\mathcal{Q}_k = \texttt{CrossAttBlock}(\tilde{\mathcal{Q}}_{k-1}, [\tilde{\mathcal{F}}_a, \tilde{\mathcal{F}}_t])$$

where $\texttt{CrossAttBlock}(\mathbf{z}, \mathbf{u})$ is the cross-attention block with $\mathbf{z}$ generating query and $\mathbf{u}$ key/value.

Finally, with $\mathcal{Q}_K$ after the $K^{\text{th}}$ layer in decoder, we use a spatial head, consisting of an MLP module, to predict the bounding boxes of the targets via $\mathcal{B} = \texttt{SpatialHead}(\mathcal{Q}_K)$, where $\mathcal{B} \in \mathbb{R}^{N_v \times N_q \times D_b}$ and $D_b = 4$ is the central position, width and height of predicted box. In addition, inspired by MDETR (Kamath et al., 2021), we predict the index for each bounding box, which corresponds to positional index of words in the original text, and is used to determine the class of each bounding box, via $\mathcal{G} = \texttt{ClsHead}(\mathcal{Q}_K)$, where $\texttt{ClsHead}(\cdot)$ is a MLP module and $\mathcal{G} \in \mathbb{R}^{N_v \times N_q \times N_t}$ with $N_t$ denoting the maximum positional indexes for any given sentence.

### 4.2.2 TEMPORAL DECODER FOR GROUNDING.

**Text-guided Temporal Query Generation.** Temporal decoder predicts start and end timestamps. Similar to spatial decoder, we use text as guidance to select target-relevant motion features to produce the initial query $\mathcal{P}_0$ as follows,

$$\mathcal{P}_0 = \{p_i^0\}_{i=1}^{N_v}, \quad p_i^0 = \texttt{AvgPooling}(\texttt{Top}_\texttt{M}(\texttt{Sim}(\tilde{\mathcal{F}}_m, \bar{\mathcal{F}}_t)))$$

where $p_i^0$ is query feature in frame $i$. $\bar{\mathcal{F}}_t$ is the pooled textual feature and $\tilde{\mathcal{F}}_m$ the motion feature extracted from $\tilde{\mathcal{F}}$. It is worth noting that, in OmniSTVG, all targets share the same start and end timestamps with the textual expression. Thus, each frame $i$ is assigned with a single initial query $p_i^0$.

**Temporal Decoding.** In temporal decoding, we send $\mathcal{P}_0$ to a decoder with $K$ layers for interaction with multimodal feature. Specifically, let $\mathcal{P}_{k-1}$=$\{p_i^{k-1}\}_{i=1}^{N_v}$ be query features fed to the $k^{\text{th}}$ ($1 \leq k \leq K$) layer for learning $\mathcal{P}_k$, where $p_i^{k-1}$ is the feature of frame $i$. To capture temporal relation, we first perform temporal attention on $\mathcal{P}_{k-1}$, as follows,

$$\{\tilde{p}_i^{k-1}\}_{i=1}^{N_v} = \texttt{TABlock}(\{p_i^{k-1}\}_{i=1}^{N_v})$$

where $\tilde{\mathcal{P}}_{k-1}=\{\tilde{p}_i^{k-1}\}_{i=1}^{N_v}$ is the query feature after temporal attention. After this, we interact $\tilde{\mathcal{P}}_{k-1}$ with the multimodal feature using cross-attention for learning $\mathcal{P}_k$, as follows,

$$\mathcal{P}_k = \texttt{CrossAttBlock}(\tilde{\mathcal{P}}_{k-1}, [\tilde{\mathcal{F}}_m, \tilde{\mathcal{F}}_t])$$

where $\tilde{\mathcal{F}}_m$ and $\tilde{\mathcal{F}}_t$ are motion and textual features from $\tilde{\mathcal{F}}$.

After $K$ layers in temporal decoder, we obtain $\mathcal{P}_K$ and employ a temporal head implemented by MLP to predict the start and end timestamps through $\mathcal{H} = \texttt{TemporalHead}(\mathcal{P}_K)$, where $\mathcal{H} \in \mathbb{R}^{N_v \times 2}$ contains the start and end probabilities $\mathcal{H}_s$ and $\mathcal{H}_e$ of $N_v$ frames for temporal localization of targets.

### 4.3 SPATIAL-TEMPORAL TUBE GENERATION

In OmniTube, we perform two steps, including *tubelet matching* and *filtering*, to generate the spatial-temporal tube for each target. **(i) Tubelet matching.** Spatial grounding predicts $N_q$ bounding boxes for each frame. To match these boxes across different frames, we apply Hungarian matching (Kuhn, 1955) using spatial positions and object class of bounding boxes, ultimately generating $N_q$ initial tubelets. **(ii) Tubelet filtering.** To further optimize tubelets, we first refine their temporal boundaries using start and end timestamps. For each tubelet, we average class probabilities of all its bounding boxes for determining the class. After that, we remove tubelets whose classes are not present in text.

### 4.4 OPTIMIZATION

In OmniTube, we predict both the spatial bounding boxes and temporal start and end timestamps for loss optimization. Due to limited space, please see our loss function in the ***supplementary material***.

## 5 EXPERIMENTS

**Implementation.** OmniTube is implemented using PyTorch (Paszke et al., 2019). We adopt ResNet-101 (He et al., 2016), VidSwin (Liu et al., 2022b), and RoBERTa (Liu et al., 2019) to extract appearance, motion, and text features. Following (Jin et al., 2022; Lin et al., 2023; Gu et al., 2024), part of parameters, including 2D/text backbones and multimodal encoder, are initialized using pre-trained MDETR (Kamath et al., 2021). We train OmniTube end-to-end, keeping the 3D backbone frozen while training all other parameters. During training, we use the Adam optimizer (Kingma & Ba, 2014) with an initial learning rate of $1e-5$ for backbone and $1e-4$ for remaining modules. Additionally, data augmentations such as random resizing and cropping are applied to all training videos, with the shorter side resized to 320 pixels. The video length $N_v$ depends on the video duration, with frames extracted at FPS=2, and the text length $N_t$ is set to 30. The channel dimensions $D_a$, $D_m$, $D_t$, and $D$ are set to 2,048, 768, 768, and 256. The $\lambda_h$ and $\lambda_k$ are set to 2 and 1.

Table 3: Comparison with current approaches. † indicates algorithm adapted for OmniSTVG.

| Methods | m_tIoU | m_vIoU | vIoU@0.3 | vIoU@0.5 |
|---|---|---|---|---|
| **(a) BOSTVG$_{Tst}$-Low** | | | | |
| TubeDETR† | 31.20 | 7.99 | 3.79 | 0.21 |
| STCAT† | 33.68 | 8.52 | 4.03 | 0.38 |
| CG-STVG† | 32.70 | 8.29 | 4.22 | 0.32 |
| OmniTube | **36.16** | **10.11** | **7.16** | **1.09** |
| **(b) BOSTVG$_{Tst}$-Medium** | | | | |
| TubeDETR† | 30.40 | 5.81 | 0.00 | 0.00 |
| STCAT† | 31.72 | 6.20 | 0.00 | 0.00 |
| CG-STVG† | 29.70 | 5.30 | 0.37 | 0.00 |
| OmniTube | **34.89** | **7.24** | **1.85** | 0.00 |
| **(c) BOSTVG$_{Tst}$-High** | | | | |
| TubeDETR† | 30.27 | 3.91 | 0.00 | 0.00 |
| STCAT† | 31.56 | 4.36 | **1.37** | 0.00 |
| CG-STVG† | 28.09 | 3.22 | **1.37** | 0.00 |
| OmniTube | **32.27** | **4.42** | **1.37** | 0.00 |
| **(d) BOSTVG$_{Tst}$-Full** | | | | |
| TubeDETR† | 31.05 | 7.52 | 3.14 | 0.17 |
| STCAT† | 33.31 | 8.03 | 3.35 | 0.31 |
| CG-STVG† | 32.09 | 7.66 | 3.56 | 0.26 |
| OmniTube | **35.83** | **9.47** | **6.17** | **0.89** |

### 5.1 STATE-OF-THE-ART COMPARISON

Since there are no available approaches specially designed for our OmniSTVG task, we adapt three STVG frameworks with source codes, including TubeDETR (Yang et al., 2022), STCAT (Jin et al., 2022), and CG-STVG (Gu et al., 2024), with appropriate modifications to their implementations, and compare our OmniTube to these approaches on the BOSTVG$_{Tst}$. Please ***notice*** that, all methods in comparison are trained on BOSTVG$_{Tra}$ for fairness, and their codes will be released.

Tab. 3 demonstrates the results and comparison of OmniTube with other methods. We can see that OmniTube outperforms other methods on all metrics. Specifically, in the full BOSTVG$_{Tst}$, OmniTube achieves 35.83% m_tIoU and 9.47% m_vIoU scores, which largely surpasses CG-STVG with 32.09% m_tIoU and 7.66% m_vIoU scores, STCAT with 33.31% m_tIoU and 8.03% m_vIoU scores, and TubeDETR with 31.05% m_tIoU and 7.52% m_vIoU scores, showing superiority. In addition, under other settings with low-, medium-, and high-density, OmniTube largely surpasses other models, showing its consistent advantages.

## 5.2 ABLATION STUDY AND DISCUSSION

**Ablation of spatial decoder.** To study different modules in spatial decoder, we conduct an ablation in Tab. 4. In our spatial decoder, the text-guided spatial query generation (**TG-SQG**) aims to learn target-specific spatial query for better interaction with multimodal feature. The spatial attention block (**SAB**) and temporal attention block (**TAB**) are applied to model spatial and temporal context

Table 4: Ablation of spatial decoder.

| | TG-SQG | SAB | TAB | m_tIoU | m_vIoU |
|---|---|---|---|---|---|
| ❶ | - | - | - | 34.33 | 8.25 |
| ❷ | - | ✓ | ✓ | 34.13 | 9.00 |
| ❸ | ✓ | - | ✓ | 34.98 | 8.89 |
| ❹ | ✓ | ✓ | - | 35.42 | 9.15 |
| ❺ | ✓ | ✓ | ✓ | **35.83** | **9.47** |

within the video. As in Tab. 4, without TG-SQG, SAB and TAB, the m_tIoU and m_vIoU scores are 34.33% and 8.25% (❶). When SAB and TAB are added, the model better captures crucial spatial and temporal cues across frames, achieving comparable m_tIoU of 34.13% but higher m_vIoU of 9.00% (❶ *v.s.* ❷). This suggests that the spatial and temporal models in videos are crucial for target localization. More importantly, as shown in Tab. 4, when incorporating TG-SQG with either SAB or TAB, both m_tIoU and m_vIoU are improved (❶ *v.s.* ❸ and ❶ *v.s.* ❹). This indicates that the text-guided spatial query contains more discriminative target-specific inforamtion and is able to better interact with and learn from the multimodal features for target localization. Finally, when all modules are applied, we obtain the best performance with with 35.83% m_tIoU and 9.47% m_vIoU scores (❺), which validates the effectiveness of our decoder design in modeling spatial and temporal cues and learning target-specific query feature for improving grounding performance.

**Ablation of temporal decoder.** To analyze temporal decoder, we conduct an ablation in Tab. 5. In our temporal decoder, the text-guided temporal query generation (**TG-TQG**) is designed to generate target-aware temporal query for localization, and the temporal attention block (**TAB**) is adopted to capture temporal relation across frames. As shown in Tab. 5, without using TG-TQG and TAB, the m_tIoU and m_vIoU scores are

Table 5: Ablation of temporal decoder.

| | TG-TQG | TAB | m_tIoU | m_vIoU |
|---|---|---|---|---|
| ❶ | - | - | 26.06 | 6.66 |
| ❷ | - | ✓ | 35.00 | 8.98 |
| ❸ | ✓ | - | 26.00 | 6.82 |
| ❹ | ✓ | ✓ | **35.83** | **9.47** |

26.06% and 6.66% (❶). When adding TAB for temporal modeling, we can see that the m_tIoU and m_vIoU scores are significantly improved to 35.00% and 8.98% (❶ *v.s.* ❷). This indicates that modeling temporal dependencies with TAB is essential for accurate temporal localization. When using TG-TQG alone, we achieve the similar m_tIoU score of 26.00% and m_vIoU score of 6.82% (❶ *v.s.* ❸). This suggests that although TG-TQG provides more discriminative target-aware cues, its benefit cannot be fully realized without temporal modeling. When combining TG-TQG and TAB, we obtain the best 35.83% m_tIoU and 9.47% m_vIoU scores (❹). This demonstrates that TG-TQG and TAB play complementary roles and their synergy leads to more accurate spatio-temporal grounding.

**Ablation of different class predictions.** In OmniTube, rather than directly predicting the bounding-box class for tube generation, we predict the position index of the class in the text as the bounding box class. To compare these

Table 6: Ablation of box class.

| | Classification | m_tIoU | m_vIoU |
|---|---|---|---|
| ❶ | Box Class | 35.36 | 8.82 |
| ❷ | Position Index | **35.83** | **9.47** |

two strategies, we conduct an ablation in Tab. 6. As shown in Tab. 6, predicting the position index yields consistently better performance (❶ *v.s.* ❷). We argue the reason is that, directly selecting from 287 classes is a challenging high-cardinality classification problem, predicting the position index simplifies category localization by linking each box to its corresponding noun phrase in the text, therefby leading to better performance.

**Ablation of parameter** $M$ **in the decoder.** In the decoder, $M$ controls how many video features are selected as relevant to the textual query. When $M$ is too small, the selected video features are insufficient to capture target-specific information from videos, which may results in

Table 7: Ablation of $M$.

| | | m_tIoU | m_vIoU |
|---|---|---|---|
| ❶ | $M = 2$ | 35.30 | 9.14 |
| ❷ | $M = 5$ | **35.83** | **9.47** |
| ❸ | $M = 10$ | 35.45 | 9.34 |

Table 8: Ablation of training data.

| | Training Data | # Objects | # Classes | # Size | m_tIoU | m_vIoU |
|---|---|---|---|---|---|---|
| ❶ | BOSTVG$_{Tra}$ | 1-10 | 287 | 8K | **35.83** | **9.47** |
| ❷ | HCSTVG-v2 (training set) | 1 | 1 | 10K | 22.24 | 4.76 |
| ❸ | BOSTVG$_{Tra}$ + HCSTVG-v2 (training set) | 1-10 | 287 | 18K | 35.47 | 9.31 |

performance degeneration. Conversely, when $M$ is too large, the selected video features may contain noise, leading to suboptimal performance. To study the effect of $M$, we conduct an ablation in Tab. 7. As shown in Tab. 7, we can observe that the best result is obtained when $M$ is 5 (❷), indicating that five query-relevant video features are sufficient for optimal grounding.

**Ablation of training data.** OmniTube is trained with our BOSTVG$_{Tra}$. To examine whether incorporating existing datasets can further improve performance, we evaluate the impact of adding existing HCSTVG-v2 with single-object data for training. The results are reported in Tab. 8. As shown in Tab. 8, training with our multi-object BOSTVG$_{Tra}$ alone achieves the best results (❶), indicating that multi-object supervision by BOSTVG$_{Tra}$ is essential for OmniTube. In contrast, training solely with HCSTVG-v2 leads to the lowest performance (❷), because it lacks multi-object interactions and provides a weaker supervisory signal for the spatial–temporal grounding of multiple targets. When combining BOSTVG$_{Tra}$ and HCSTVG-v2 for training, the model performance slightly decreases (❸). This degradation is likely due to inconsistent training distributions for different tasks. HCSTVG-v2 contains only one human class and lacks multi-object queries. When merging BOSTVG$_{Tra}$ and HCSTVG-v2, the model is exposed to a large amount of samples with a very narrow semantic scope, which may bias learning toward human-centric patterns and weaken its ability to generalize across diverse object categories. Moreover, since HCSTVG-v2 lacks multi-object queries, it cannot provide supervision for joint localization of multiple entities. These two factors together result in the final slight performance drop, which also indicates that *training on diverse, multi-class, multi-object data* is essential for solving the omni-target problem. We believe that this result underscores the importance of our dataset design: BOSTVG is not just an extension of existing work, but a necessary shift toward richer, more complex training signals.

**Impact of motion information.** In OmniTube, we leverage motion information as an additional cue for target localization. To evaluate its effect, we conduct an ablation in Tab. 9. As shown in Tab. 9, incorporating motion feature consis-

Table 9: Ablation of the motion cue.

| | Motion | m_tIoU | m_vIoU |
|---|---|---|---|
| ❶ | OmniTube w/o motion | 35.29 | 8.88 |
| ❷ | OmniTube w/ motion | **35.83** | **9.47** |

tently improves localization performance (❶ *v.s.* ❷). This is because motion encodes strong temporal signals that help link entities across frames and clarify the interactions among targets. Without motion cues, the model depends solely on appearance, which makes it harder to capture dynamic relationships and maintain temporal consistency. The results in Tab. 9 confirm that motion features play a complementary role to appearance cues and are essential for spatio-temporal grounding.

Due to space limitations, we show additional results and analyzes in the ***supplementary material***.

## 6  CONCLUSION AND ETHICAL STATEMENT

In this work, we present OmniSTVG that aims to locate all targets in the query. To foster research on OmniSTVG, we propose BOSTVG by offering 10,018 sequences with 10.2 million frames. To our best knowledge, BOSTVG is the first benchmark dedicated to OmniSTVG. Moreover, to encourage future research, we introduce OmniTube, a simple yet highly effective method for OmniSTVG. Our results show the advantages of OmniTube over other models.

**Ethical Statement.** The construction of BOSTVG strictly adheres to ethical standards, ensuring that all annotators provided informed consent. All videos in BOSTVG are collected under Creative Commons licenses and used for research purpose only. Due to limited space, we provide more detailed ethical statement in the ***supplementary material***.

**Acknowledgments.** Libo Zhang is supported by National Natural Science Foundation of China (No. 62476266). Zhipeng Zhang is supported by Natural Science Foundation of China (No. 62503323). Bing Fan, Yan Huang, and Heng Fan are not supported by any funds for this work.

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

## SUPPLEMENTAL MATERIAL

To better understanding of this work, we provide additional details, analysis, results, and discussions:

- **A    Details of Object Categories**
  We present detailed object categories in the proposed BOSTVG, along with the number of sequences in each category.

- **B    Construction Pipeline for BOSTVG and Additional Annotation Examples**
  This section introduces the detailed construction pipeline of BOSTVG and showcases more annotation examples.

- **C    Detailed Architectures of Modules**
  We show architectures for `SAEncoder(·)`, `SABlock(·)`, and `TABlock(·)` in the paper.

- **D    Detailed Architecture of Baseline**
  We show the detailed architecture of our baseline.

- **E    Details of Evaluation Metrics**
  We describe evaluation metrics used for assessing different approaches.

- **F    Details of Loss Function for Optimization**
  We provide details of the loss function for optimization in our method.

- **G    Additional Results and Analysis**
  In this section, we provide additional comparison results and analysis of our work.

- **H    Limitation of OmniTube**
  We discuss the limitations of our proposed OmniTube.

- **I    Ethical Statement and Dataset Specification**
  We discuss the ethical concerns, annotator protection, and specifications of BOSTVG.

## A    DETAILS OF OBJECT CATEGORIES

BOSTVG contains 287 object categories, aiming to provide a diverse platform for the OmniSTVG task. The categories are organized hierarchically to ensure comprehensive coverage. Specifically, we first collect 23 coarse object classes, comprising "*Appliance*", "*Bird*", "*Body Part*", "*Building*", "*Clothes*", "*Container*", "*Environment Element*", "*Fish*", "*Food*", "*Furniture*", "*Geometric*", "*Human*", "*Invertebrate Animal*", "*Item*", "*Kitchenware*", "*Machine*", "*Mammal*", "*Music Instrument*", "*Plant*", "*Reptile Animal*", "*Sport Equipment*", "*Tool*", and "*Vehicle*". Please note, since "*Human*" is a special category, we separate it from "*Mammal*". After this, we further gather 287 fine categories from coarse classes. Fig. 4 shows the category organization of BOSTVG (**please zoom in**). We will provide and release the category information with our BOSTVG on our website.

## B    CONSTRUCTION PIPELINE FOR BOSTVG AND ADDITIONAL ANNOTATION EXAMPLES

The construction of BOSTVG comprises five steps: **(i) *Overall design***. In the first step, we determine the overall goal, all object categories to include in BOSTVG, and strategies to search for videos from YouTube. **(ii) *Video collection from YouTube***. The second step is to collect videos using strategies in the first step. In our work, we choose to source videos from YouTube because it is currently the largest video platform and contains many real-world videos. Please note that, all videos are collected under the *Creative Commons* license for *research purpose only*. **(iii) *Video selection***. After video collection, the third step is to select videos which are qualified for our OmniSTVG task. Specifically, our experts (*e.g.*, students working on related fields) will manually inspect each video for ensure all selected videos meet the requirement. **(iv) *Initial labeling***. The fourth step is to conduct the initial data labeling by our experts. Specifically, the experts will first provide a textual query for each video. Based on the query, two types of annotations will be provided, including start and end timestamps of the temporal segment and the spatial bounding boxes for the targets of interest. **(v) *Multi-round inspection and refinement***. In the final fifth step, multiple rounds of inspection and refinement if needed are performed by our experts and volunteers to ensure high-quality annotations. Fig. 5 shows the construction pipeline of BOSTVG. In Fig. 6, we provide more annotation examples in BOSTVG.

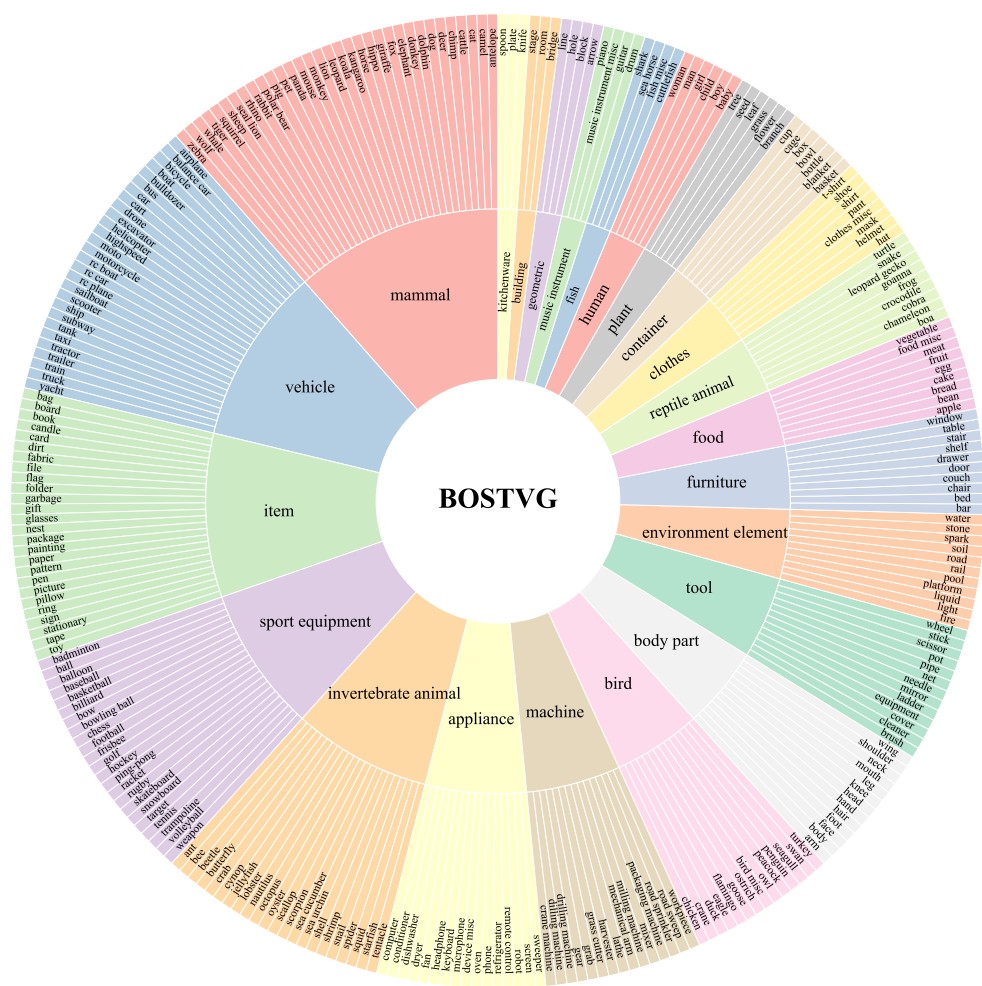

Figure 4: Category organization of BOSTVG. The inner circle of the pie chart displays 23 coarser object classes, while the outer circle displays 287 fine object categories. *Best viewed in pdf and by zooming in.*

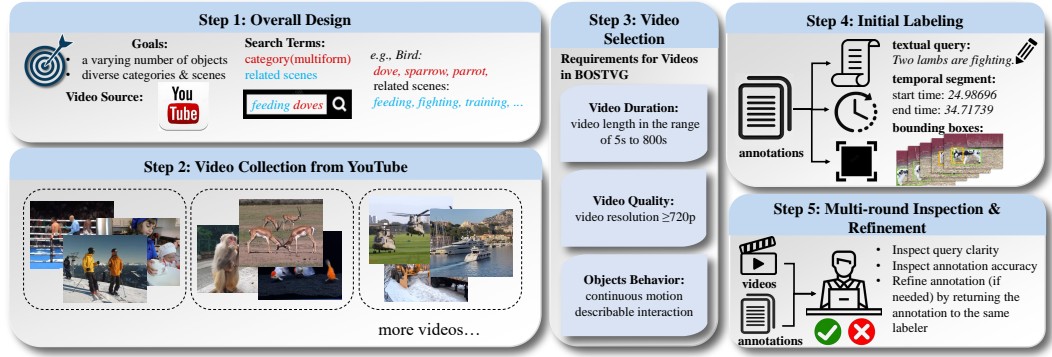

Figure 5: Illustration of construction pipeline of the proposed BOSTVG, including five key steps, *i.e.*, (i) overall design, (ii) video collection from YouTube, (iii) video selection, (iv) initial labeling, and (v) multi-round inspection and refinement.

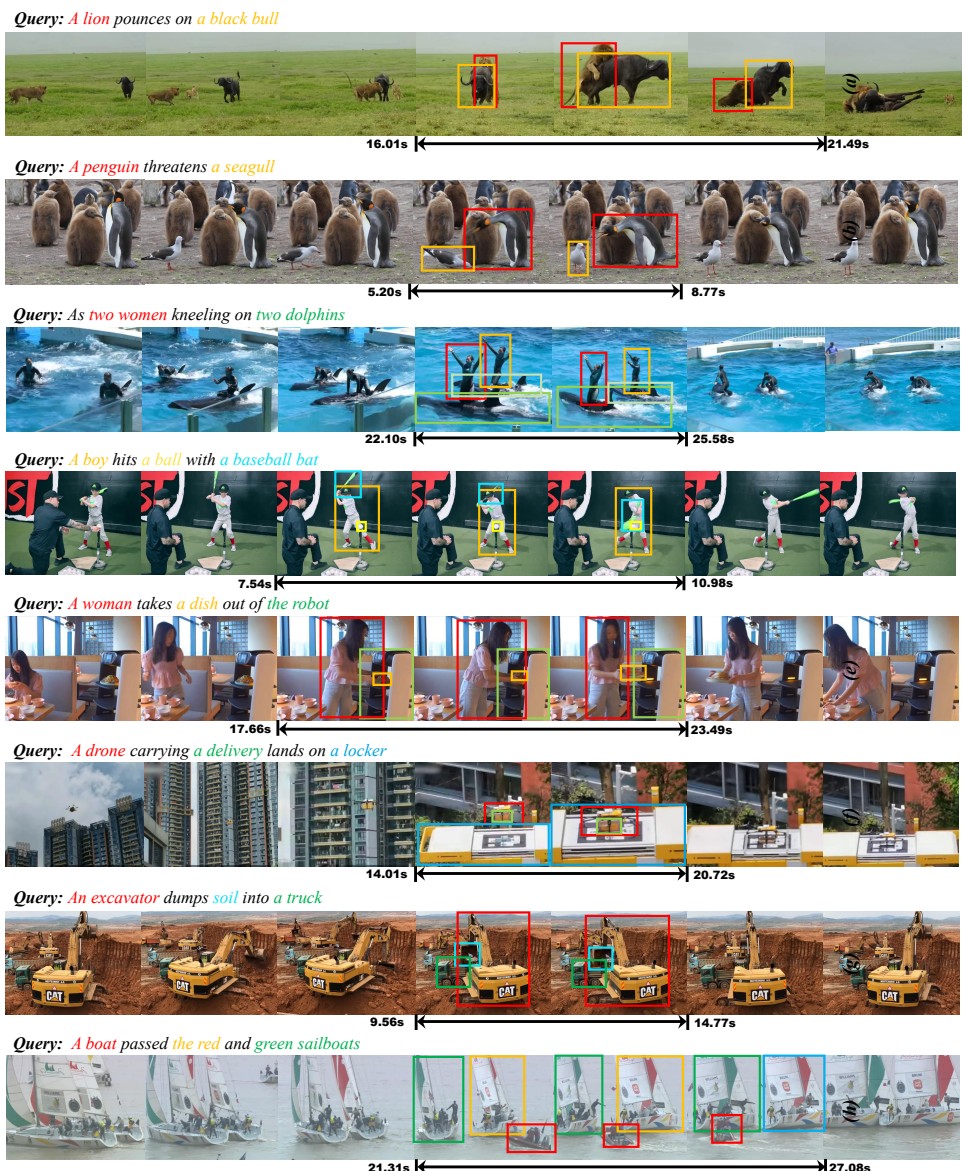

Figure 6: Additional annotation samples on our BOSTVG.

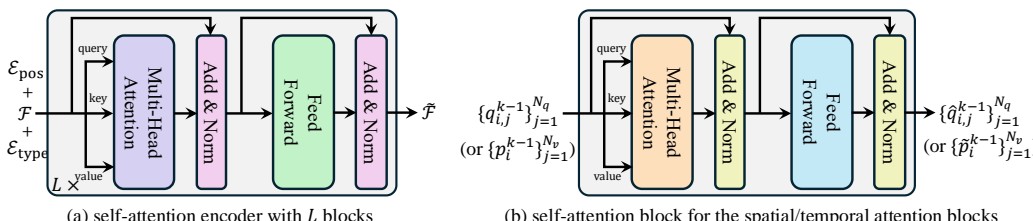

(a) self-attention encoder with $L$ blocks

(b) self-attention block for the spatial/temporal attention blocks

Figure 7: Detailed architectures for the self-attention encoder in (a) and spatial/temporal attention blocks in (b).

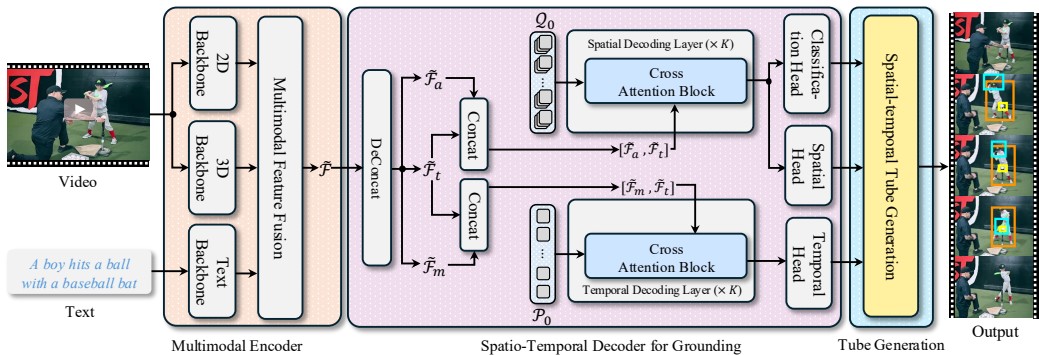

Figure 8: Illustration of our baseline, which share similar architecture with OmniTube but without text-guided query generation modules, spatial and temporal attention blocks.

## C DETAILED ARCHITECTURES OF MODULES

In Fig. 7 (a), we show the architectures for the self-attention encoder, which is composed of $L$ ($L$ = 6) standard self-attention encoder blocks and used to fuse features from multiple modalities. In Fig. 7 (b) we display the architectures for the spatial attention block and temporal attention block, which are both implemented with a self-attention block.

## D DETAILED ARCHITECTURE OF BASELINE

Our baseline method mentioned in the main text shares a similar architecture with OmniTube but does not contain the key text-guided query generation modules and spatial and temporal attention blocks. Its detailed architecture is shown in Fig. 8. From the experiments described in the main text, the performance of the baseline is inferior to our OmniTube, which validates the effectiveness of our text-guided query generation and spatial and temporal attention blocks for target localization.

## E DETAILS OF EVALUATION METRICS

Following current STVG benchmarks (Chen et al., 2019; Zhang et al., 2020b), we utilize multiple metrics, including $m\_tIoU$, $m\_vIoU$, and $vIoU@R$ for evaluation. Specifically, $m\_tIoU$ aims to assess the temporal localization performance and is calculated by averaging the temporal IoU scores $tIoU$ on all test videos. The $tIoU$ is calculated as $\frac{|\mathcal{P}_i|}{|\mathcal{P}_u|}$, where $\mathcal{P}_i$ and $\mathcal{P}_u$ represent the intersection and union between the groundtruth and predicted segments, respectively. $m\_vIoU$ is utilized to measure the spatial localization performance, which is calculated by averaging spatial IoU scores $vIoU$. The $vIoU$ is calculated as follows,

$$vIoU = \frac{1}{|\mathcal{P}_u|} \sum_{t \in \mathcal{P}_i} \left( \frac{1}{N_q} \sum_{i \in N_q} \texttt{IoU}(b_{t,i}^*, b_{t,i}) \right)$$

where $b_{t,i}^*$ and $b_{t,i}$ are the groundtruth bounding box and the predicted bounding box of the $i^{\text{th}}$ target in $t^{\text{th}}$ frame. The $vIoU@R$ is defined as ratio of samples with spatial IoU scores above threshold $R$.

## F DETAILS OF LOSS FUNCTION FOR OPTIMIZATION

Given a video and its textual expression, OmniTube predicts two types of localization, comprising (1) the spatial position $\mathcal{B}$ and class $\mathcal{G}$ of the bounding box in the spatial grounding, and (2) the start timestamps $\mathcal{H}_s$ and end timestamps $\mathcal{H}_e$ in the temporal grounding. During training, given the ground truth for the spatial location $\mathcal{B}^*$ and class $\mathcal{G}^*$ of the bounding box, as well as the start timestamps $\mathcal{H}_s^*$ and end timestamps, we can calculate the total loss as

$$\mathcal{L} = \lambda_h \mathcal{L}_h((\mathcal{B}, \mathcal{G}), (\mathcal{B}^*, \mathcal{G}^*)) + \lambda_k (\mathcal{L}_k(\mathcal{H}_s^*, \mathcal{H}_s) + \mathcal{L}_k(\mathcal{H}_e^*, \mathcal{H}_e))$$

where $\mathcal{L}_h$ denotes the Hungarian loss in (Carion et al., 2020) as explained later and $\mathcal{L}_k$ represents the KL divergence loss. The parameters $\lambda_h$ and $\lambda_k$ are used to balance the loss.

**Explanation of $\mathcal{L}_h$.** In spatial grounding, we predict the spatial location and class of the bounding box, which can be denoted as

$$\mathcal{Y} = (\mathcal{B}, \mathcal{G}) = \{\{y_{i,j}\}_{i=1}^{N}\}_{j=1}^{N_q}$$

where $N$ and $N_q$ denote the number of frames and the number of predicted targets, respectively. $y_{i,j} = (b_{i,j}, g_{i,j})$, where $b_{i,j}$ and $g_{i,j}$ are the spatial location and class of the $j^{\text{th}}$ bounding box in the $i^{\text{th}}$ frame. Suppose that the ground truth for spatial location and class can be denoted as

$$\mathcal{Y}^* = (\mathcal{B}^*, \mathcal{G}^*) = \{\{y_{i,j}^*\}_{i=1}^{N}\}_{j=1}^{N_q^*}$$

where $N_q^*$ ($\leq N_q$) denotes the number of ground truth targets. Following DETR (Carion et al., 2020), to compute the pair-wise matching cost between the prediction results and ground truth, we first utilize the Hungarian matching algorithm to establish a one-to-one correspondence between the prediction and ground truth, as follows,

$$\hat{y} = \texttt{HungarianMatcher}(\mathcal{Y}^*, \mathcal{Y})$$

where $\hat{y}$ represents the successfully matched bounding boxes, while the unmatched bounding boxes are excluded from the loss calculation. After this, we can calculate the loss $\mathcal{L}_h$ as follows,

$$\mathcal{L}_h = \lambda_u \mathcal{L}_u(\mathcal{B}^*, \hat{\mathcal{B}}^*) + \lambda_l \mathcal{L}_l(\mathcal{B}^*, \hat{\mathcal{B}}^*) + \lambda_c \mathcal{L}_c(\mathcal{G}^*, \hat{\mathcal{G}}^*)$$

where $\mathcal{L}_u$, $\mathcal{L}_l$ and $\mathcal{L}_c$ are IoU, smooth L1, and binary cross-entropy losses. The parameters $\lambda_u$, $\lambda_l$, $\lambda_c$ are set to 3, 5, 1.

# G ADDITIONAL RESULTS AND ANALYSIS

## G.1 COMPARISON WITH EXISTING METHODS ON CLASSIC STVG BENCHMARK

Besides OmniSTVG, OmniTube can also be used for conventional single-object STVG. In order to validate this, we conduct an experiment on the classic single-object STVG benchmark HCSTVG-v2 (Tang et al., 2021). Tab. 10 shows the results. Since OmniTube focuses on multi-object localization, including multi-object interaction and multi-object tube matching, it does not achieve new state-of-the-art on the single-object STVG benchmark. That said, our method outperforms most existing approaches, showing its effectiveness.

Table 10: Comparison with existing STVG methods on the classic STVG dataset HCSTVG-v2.

| Methods | m_tIoU | m_vIoU | vIoU@0.3 | vIoU@0.5 |
|---|---|---|---|---|
| MMN (Lei et al., 2020) | - | 30.3 | 49.0 | 25.6 |
| TubeDETR (Yang et al., 2022) | 53.9 | 36.4 | 58.8 | 30.6 |
| STCAT (Jin et al., 2022) | 56.6 | 36.9 | 60.3 | 33.6 |
| STVGFormer (Lin et al., 2023) | 58.1 | 38.7 | 65.5 | 33.8 |
| CG-STVG (Gu et al., 2024) | 60.0 | 39.5 | 64.5 | 36.3 |
| OmniTube | 58.5 | 38.4 | 63.1 | 33.4 |

## G.2 COMPARISON WITH SINGLE-OBJECT STVG METHOD ON BOSTVG

As discussed earlier in the main text, to adapt existing STVG to OmniSTVG, a straightforward solution is to employ the STVG model repeatedly for multiple single-object textual queries. Here we conduct an experiment by repeatedly running a recent STVG method CG-STVG (Gu et al., 2024) trained on Vid-STG (Zhang et al., 2020b) and comparing it

Table 11: Comparison with the single-object STVG method CG-STVG on OmniSTVG. Please note that the results here are reported on a newly built test set from BOSTVG$_{\text{Tst}}$ with 283 videos.

| Methods | m_tIoU | m_vIoU | vIoU@0.3 | vIoU@0.5 | Time Cost |
|---|---|---|---|---|---|
| CG-STVG | 32.5 | 8.6 | 3.9 | 0.3 | 558.1s |
| OmniTube | 37.9 | 10.5 | 6.4 | 0.4 | 134.1s |

with OmniTube. Specifically, we first build a new test set by selecting videos from BOSTVG$_{\text{Tst}}$ that have same classes as in VidSTG so that CG-STVG can work. There are 283 sequences selected. Then we provide a textual query for each single object in these videos so CG-STVG can successfully run. Tab. 11 reports the results. As in Tab. 11, our OmniTube achieves better results due to multi-object relationship modeling and runs much faster without the need of running multiple times.

### G.3  COMPARISON WITH EXISTING MLLM-BASED METHODS ON BOSTVG

To investigate the STVG capabilities of multimodal large models, we have added experiments evaluating two popular MLLMs, including Gemini-2.5 Pro (Comanici et al., 2025) and Qwen2.5-VL (Bai et al., 2025), on OmniSTVG. Please note that, since Gemini-2.5 Pro is not open-source, we directly apply Gemini-2.5 Pro to predict spatial-temporal locations of targets in videos without any tuning. For Qwen2.5-VL, we report results of two versions, include zero-shot version (named Qwen2.5-VL) without tuning and supervised fine-tuned version (named Qwen2.5-VL-SFT). The results and comparison on BOSTVG$_{\text{Tst}}$-Full are shown in Tab. 12. From Tab. 12, we observe that, our proposed OmniTube (❹) outperforms all MLLM-based methods (❶: Gemini-2.5 Pro, ❷: Qwen2.5-VL without tuning, ❸: Qwen2.5-VL-SFT with supervised fine-tuning). We argue that the reasons why MLLM-based methods underperform are two fold: (1) without task-specific tuning (see ❶ and ❷), MLLMs-based method are not equipped to perform fine-grained spatio-temporal localization for multiple objects in videos; and (2) even with fine-tuning (see ❸), the training objective of MLLM treats structured outputs such as timestamps and bounding boxes as discrete text tokens, making them particularly sensitive to even small lexical differences (e.g., "1–9s" versus "0–8s"). These findings highlight the challenges of our OmniSTVG task and the necessity of dedicated architectures like the proposed OmniTube.

Table 12: Comparison with existing mllm-based methods on BOSTVG$_{\text{Tst}}$-Full.

| Method | m_tIoU | m_vIoU | vIoU@0.3 | vIoU@0.5 |
|---|---|---|---|---|
| ❶ Gemini-2.5 Pro | 35.5 | 7.2 | 5.1 | 0.5 |
| ❷ Qwen2.5-VL | 15.2 | 1.6 | 0.0 | 0.0 |
| ❸ Qwen2.5-VL-SFT | 29.1 | 5.2 | 2.2 | 0.1 |
| ❹ OmniTube (ours) | **35.8** | **9.5** | **6.2** | **0.9** |

### G.4  ZERO-SHOT GROUNDING PERFORMANCE COMPARISON

To evaluate the zero-shot object grounding capability of different methods, we conduct a new zero-shot experiment by splitting BOSTVG into non-overlapped training and test sets. Specifically, the training set contains 237 classes, and the test set contains the rest 50 held-out classes, with no overlap between training and test sets. As shown in Tab. 13, both our OmniTube and existing methods are degraded to some extend (see Tab. 13 here and Tab. 3 (d) in the paper) in this zero-shot setting due to lack of training samples for unseen cases. That being said, the proposed OmniTube still achieves superior performance than existing methods, which validates the effectiveness and generality of our architecture in grounding zero-shot objects.

Table 13: Comparison of zero-shot grounding ability of different methods.

| Method | #Train Classes | #Test Classes | m_tIoU | m_vIoU | vIoU@0.3 | vIoU@0.5 |
|---|---|---|---|---|---|---|
| ❶ STCAT | 237 | 50 | 28.71 | 6.92 | 2.75 | 0.25 |
| ❷ CG-STVG | 237 | 50 | 29.84 | 7.24 | 2.98 | 0.26 |
| ❸ OmniTube (ours) | 237 | 50 | **30.98** | **7.58** | **3.56** | **0.37** |

### G.5  ABLATION OF DIFFERENT FEATURES

To explore the role of different features, we show the experiments by studying the ResNet and VidSwin feature for performance in Tab.. 14. As shown in Tab. 14, when using ResNet feature only (*i.e.*, removing VidSwin feature; see ❶), the m_tIoU and m_vIoU are 35.29% and 8.88%, respectively, indicating that the ResNet-based appearance information is essential for grounding. This is expected because the ResNet-based appearance backbone is initialized from a pre-trained vision-language model, MDETR (Kamath et al., 2021), and provides strong object-level semantics needed for target localization. In contrast, when using VidSwin feature only (i.e., removing ResNet feature; see ❷), the model fails to converge, and therefore no valid results can be reported. This indicates that, when using only VidSwin feature, the model loses its ability to ground textual references to visual regions and fails to converge meaningfully. When combining ResNet and VidSwin features, the model achieves the best performance with 35.83% m_tIoU and 9.47% m_vIoU, demonstrating that motion cues from VidSwin complement appearance information from ResNet and further enhance target localization.

### G.6  DISCUSSION OF TASK INHERENT DIFFICULTIES.

Table 14: Ablation of different features. "×" means that the model fails to converge.

| | Appearance Feature (ResNet) | Motion Feature (VidSwin) | m_tIoU | m_vIoU |
|---|---|---|---|---|
| ❶ | ✓ | | 35.29 | 8.88 |
| ❷ | | ✓ | × | × |
| ❸ | ✓ | ✓ | 35.83 | 9.47 |

For the proposed BOSTVG benchmark, several inherent difficulties that go beyond conventional single-object STVG are introduced, including *(i) multi-object grounding*: In our benchmark and task, the query often refers to multiple objects. To localize all the mentioned objects in the query, the grounding model needs to understand the complex relationships among multiple objects for accurate localization, which is inherently challenging; *(ii) dynamic and crowded scenes:* Compared to conventional single-object STVG benchmark, our proposed benchmark brings in more frequent and severe appearance changes, e.g., motion blur, partial or full occlusions among objects, and close object-interactions in videos, thereby demanding precise spatio-temporal alignment under visual ambiguity, particularly in the context of multi-object grounding; and *(iii) more complex textual query:* In our benchmark, the textual query is

*Query: A goalkeeper is saving the soccer ball from two players, one wearing a green vest and the other with dark skin.*

*Query: The remaining four athletes standing by jumped into the water.*

*Query: An adult goose went into the water with five goslings, while another stayed on the grass.*

Figure 9: Examples illustrating the inherent challenges of the BOSTVG benchmark.

more complicated by containing more entities, due to its aim at localizing multiple objects. For robust performance, the grounding model needs to be equipped with strong scene-level reasoning.

Critically, these challenges co-occur in real-world scenarios, as illustrated in Fig. 9. In the soccer scene (Top), the query "one wearing a green vest, the other with dark skin" refers to two players among many, requiring the model to ground multiple targets amid crowd clutter, motion blur, and partial occlusion while correctly parsing fine-grained visual attributes from a linguistically complex description. The diving sequence (Middle) involves four athletes jumping simultaneously. The query must be interpreted to identify all four individuals despite their near-identical appearance, synchronized motion, and rapid entry into the water a setting that combines dense multi-object grounding, severe visual ambiguity, and a query implicitly referencing a group action. Similarly, in the goose scene (Bottom), the model must localize an adult goose and five goslings, which move into the water while another adult goose remains on land. This requires disentangling spatially and temporally varying object states, resolving visually similar instances, and understanding relational language ("another stayed on the grass"), again entangling all three core challenges. These examples underscore that BOSTVG reflects the intertwined complexity of real-world video grounding, where multi-object reference, visual dynamics, and linguistic sophistication must be addressed jointly, making it a more realistic and demanding benchmark than prior single-target settings.

## G.7 DISCUSSION OF LIMITATIONS OF EXISTING ALGORITHMS ON BOSTVG

Existing STVG methods are designed for single-object grounding. For example, the recent CG-STVG mines visual context of the object to improve localization accuracy, but it can mine only the visual context of a single object and cannot be directly extended to multi-object scenarios. Moreover, existing methods lack explicit modeling of interactions among multiple objects, making them unable to resolve ambiguities when several similar or interacting objects appear together, which is common in BOSTVG. In contrast, the proposed OmniTube is built from the ground up for omni-object grounding, supporting joint parsing, localization, and disambiguation of multiple targets.

### G.8    Discussion of Data Distribution in BOSTVG

Our BOSTVG test set is partitioned into three complexity-based groups: BOSTVG-Low (1–3 targets, 1,566 samples), BOSTVG-Medium (4–6 targets, 273 samples), and BOSTVG-High ($\geq$7 targets, 73 samples). Similar to other well-known datasets such as ImageNet (Deng et al., 2009) and COCO (Lin et al., 2014), the data distribution of our BOSTVG follows the long-tailed distribution, reflecting the natural frequency of multi-object scenarios in the real world. As a consequence, queries referring to 1–3 objects are more common, while those with $\geq$7 targets are rare but still included. To ensure fair evaluation across complexity levels, in addition to the overall average, we report separate metrics for BOSTVG-Low, -Medium, and -High (please see Tab. 3 in the paper). However, we agree that adding more samples in BOSTVG-High is crucial for evaluating the model's capability in handling complex, multi-target scenarios. Addressing this, we will include more high-density examples (*e.g.*, 200 more) in BOSTVG-High to enable more robust evaluation of scalability. Since this is beyond our current goal, we leave this to our future work.

### G.9    Discussion of the Shared Temporal Segmentation Assumption

In this work, we follow the conventional standard STVG formulation, in which the textual query describes a spatio-temporal event defined by determined start and end time stamps. The only difference in our OmniSTVG task is, the event contains multiple simultaneous interactions among multiple objects, and there all target objects share the same start and end time stamps (*i.e.*, same temporal segmentation) with the event itself. This design reflects how humans typically describe dynamic scenes: multi-object references are anchored to a single event interval. That being said, we believe more formats of spatio-temporal video grounding is worth studying, *e.g.*, multiple objects with each having its own temporal segmentation. Since this is currently beyond our goal, we leave its exploration to our future.

### G.10    Analysis of Memory Requirement and Model Efficiency of OmniTube

In this work, we train our OmniTube using 32 Nvidia A100 (48GB) for around 13 hours. In inference, for a 50s-video with 100 sampled frames, the GPU memory requirement for running OmniTube is 34GB and the inference time is 0.66 seconds.

### G.11    Qualitative Results

To further qualitatively validate the effectiveness of our OmniTube, we provide the grounding results of our method in Fig. 10. As shown in Fig. 10, we can see that, our method can robustly localize all objects mentioned in the textual query, showing its effectiveness.

## H    Limitation of OmniTube

To encourage more future research on OmniSTVG, we propose a simple yet effective baseline OmniTube. Despite achieving promising results, OmniTube has two limitations. *First*, OmniTube may degrade in highly dense scenarios. In highly dense videos, target objects may be heavily occluded, resulting in lower localization performance. To handle this case, special strategies such as leveraging rich spatial context information via explicit object-object interaction or target trajectory cues are needed for improving target localization. *Second*, our OmniTube, similar to existing STVG models, requires considerable computational resources for training because it takes as input the whole video. To alleviate this, we will investigate more light-weight resource-friendly frameworks for OmniSTVG. Considering that our current goal is to provide a feasible baseline for OmniSTVG, we leave the above explorations for better performance to our future work.

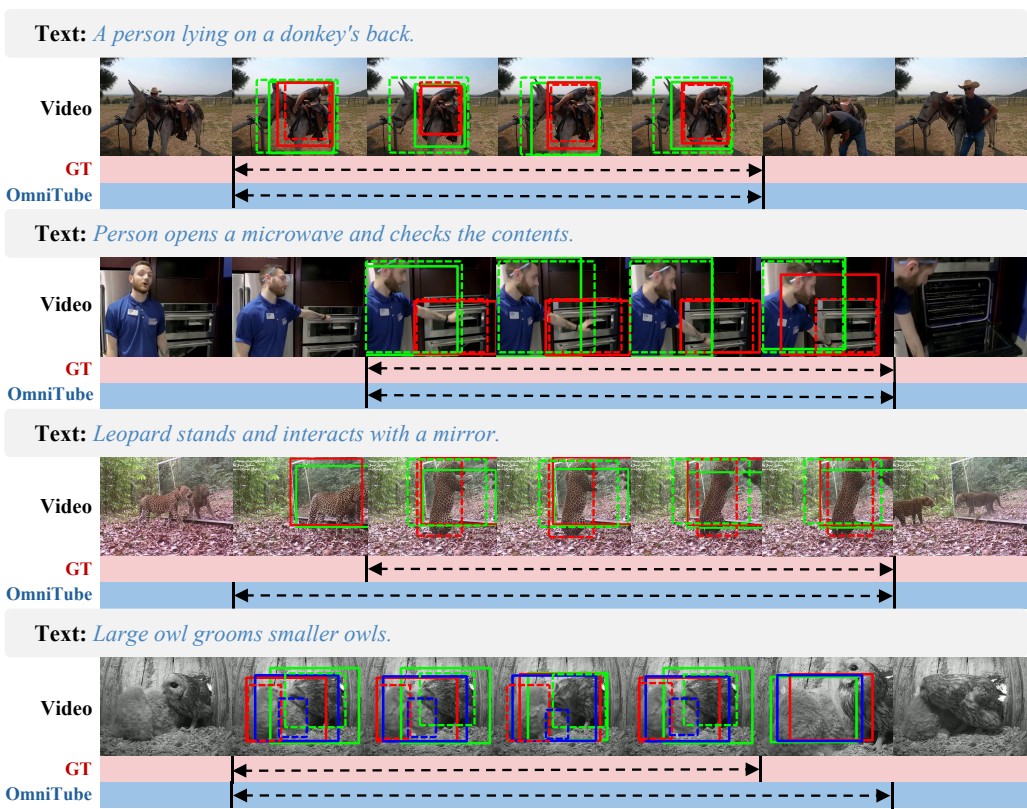

Figure 10: Qualitative results of our method. Our prediction results are visualized with ***solid-line*** bounding boxes, and the groundtruth boxes are shown in the ***same color*** with the prediction results but with ***dashed-line*** bounding boxes.

# I ETHICAL STATEMENT AND DATASET SPECIFICATION

**Ethical Concerns.** The construction of BOSTVG strictly follow ethical standards. All videos are collected under Creative Commons licenses. Nevertheless, we understand the license might change in future. Once any notification regarding this is received, we will take appropriate actions to handle it.

**Annotator Protection.** To ensure annotator well-being, all participants provide informed consent and are pre-warned about potentially distressing content. We allow annotators to skip any video that cause discomfort or to withdraw from the study entirely.

**Maintenance.** BOSTVG will be hosted on the popular Github. This allows us to conveniently check feedback from the community and to improve BOSTVG via necessary maintenance and updates by the authors. In addition, all experimental results of compared approaches on BOSTVG will be made publicly available. Our ultimate goal is to provide a long-term and stable platform for the community to foster research on spatio-temporal omni-object video grounding.

**Responsible Use of BOSTVG.** Our BOSTVG aims at fostering research and application of video grounding. BOSTVG can be used for *research purpose only*. Please note that, due to inevitable bias during the data collection process, there may exist geographic and demographic imbalance.

**Data Collection Protocol.** BOSTVG is constructed from YouTube videos licensed under Creative Commons, used strictly for research purposes. All videos and annotations undergo manual review to exclude offensive or sensitive content. During curation, we prioritize diversity in both scene types and object categories to ensure a representative and balanced dataset.

**Potential Limitations.** A key limitation of BOSTVG lies in the high cost and complexity of annotation. Precisely grounding multiple objects in videos especially under omni-target settings requires fine-grained spatiotemporal labeling which is labor-intensive time-consuming and demands expert annotators. This limits the scalability of the dataset and hinders rapid expansion to larger scales or new domains.

