# OpenReview forum: "OmniSTVG: Toward Spatio-Temporal Omni-Object Video Grounding"
_ICLR.cc/2026/Conference — ICLR 2026 Poster_

### Official Review · Reviewer_81gP · 2025-10-22

**Soundness:** 2
**Presentation:** 3
**Contribution:** 3
**Rating:** 6
**Confidence:** 3

**Summary:**

This paper introduces the OmniSTVG task where multiple objects mentioned in the text query have to be grounded spatially and temporally. The BOSTVG dataset is collected, consisting of 10K videos manually annotated with spatio-temporal tubes for 1-10 objects, including 287 object classes in total. The OmniTube model is presented, including separate spatial and temporal decoders, text-guided query generation and multi-tube prediction. This model outperforms single-object methods adapted to the new task, and is also about 4x fater than naive approaches of running a single-object model for each object in the query.

**Strengths:**

- A new dataset that addresses an important limitation of previous STVG approaches
- OmniTube performs well compared to prior approaches and has been ablated extensively

**Weaknesses:**

- Assuming all objects share the same temporal segmentation is a strong assumption
- It is confusing what the “baseline” is meant to show in the paper: it excludes various features present in prior work, e.g. text-guided query generation in https://arxiv.org/abs/2502.11168 or alternative spatial and temporal blocks in https://arxiv.org/abs/2203.16434. Note that these features are rightfully not being presented as contributions in the introduction.

**Questions:**

- Out of curiosity, is the architecture able in practice to ground “zero-shot” objects not seen in the training set?
- How much is each of ResNet and VidSwin feature helpful for the STVG performance?

**Details Of Ethics Concerns:**

A dataset has been collected hence any potential bias/responsible research practice could require more attention.

---

> ### Author Response · Authors · 2025-11-20
> **Rebuttal (part 1/2)**
>
> We thank the reviewer for the careful comments on our work and our responses are provided below.
>
> > **Q1**: Assuming all objects share the same temporal segmentation is a strong assumption.
>
>
> **A1**: Thanks for this insightful comment. In this work, we follow the conventional standard STVG formulation, in which the textual query describes a spatio-temporal event defined by determined start and end time stamps. The only difference in our OmniSTVG task is, the event contains multiple simultaneous interactions among multiple objects, and there all target objects share the same start and end time stamps (i.e., same temporal segmentation) with the event itself. This design reflects how humans typically describe dynamic scenes: multi-object references are anchored to a single event interval. That being said, we believe more formats of spatio-temporal video grounding is worth studying, _e.g._, multiple objects with each having its own temporal segmentation. Since this is currently beyond our goal, we leave its exploration to our future.
>
> Again, thanks for this valuable comment. We have incorporated the above explanation into the revised version to further clarify our task (`highlighted in red; please see L1046-L1056 in Sec. G of the supplementary material`).
>
>
>
>
> > **Q2**: It is confusing what the “baseline” is meant to show in the paper: it excludes various features present in prior work, _e.g._ text-guided query generation in [1] https://arxiv.org/abs/2502.11168 or alternative spatial and temporal blocks in [2] https://arxiv.org/abs/2203.16434. Note that these features are rightfully not being presented as contributions in the introduction.
> >
> > [1] Knowing Your Target: Target-Aware Transformer Makes Better Spatio-Temporal Video Grounding
> >
> > [2] TubeDETR: Spatio-Temporal Video Grounding with Transformers
>
> **A2**: Sorry for the confusion. The “Baseline” in Tab. 3 is *not* intended as a competing method, but more like a ablation reference to analyze the contribution of each component within our OmniTube framework. It is actually a minimal architecture for OmniSTVG (_e.g._, without text-guided query generation or advanced spatio-temporal blocks) so that we can isolate the effect of adding each module. Some of these modules are inspired by prior work (_e.g._, text-guided queries from TA-STVG [1], temporal modeling from TubeDETR [2]). However, none of these methods support multi-object grounding. Our key contribution lies in reformulating the entire pipeline for omni-object STVG, including multi-query encoding, joint tube prediction, and cross-object matching—components not present in any prior approach.
>
> To avoid confusion in this work, we revise the manuscript by removing the "baseline" in the comparision. Again, thanks for this helpful comment!
>
>
> > **Q3**: Out of curiosity, is the architecture able in practice to ground “zero-shot” objects not seen in the training set?
>
> **Tab. C: Comparison of zero-shot grounding ability of different methods.**
> |   | Method               | #Train Classes | #Test Classes | m_tIoU | m_vIoU | vIoU\@0.3 | vIoU\@0.5 |
> |:-:|----------------------|:--------------:|:-------------:|:------:|:------:|:--------:|:--------:|
> |➊  | STCAT  |      237        |      50        |  28.71 |  6.92 |  2.75   |  0.25   |
> |➋  | CG-STVG  |      237        |      50        |  29.84 |  7.24 |   2.98  | 0.26    |
> |➌ | OmniTube (ours)   |   237    |   50  |  30.98 |  7.58 |  3.56   |  0.37   |
>
> **A3**: Thanks for this insightful question. Yes, the architecture of OmniTube is able to ground "zero-shot" object not seen in the training set. As suggested, we conduct a new zero-shot experiment by splitting BOSTVG into non-overlapped training and test sets. Specificaly, the training set contains 237 classes, and the test set contains the rest 50 held-out classes, with no overlap between training and test sets. As shown in Tab. C, both our OmniTube and existing methods are degraded to some extend (see Tab. C here and Tab. 3 (d) in the paper) in this zero-shot setting due to lack of training samples for unseen cases. That being said, the proposed OmniTube still achieves superior performance than existing methods, which validates the effectiveness and generality of our architecture in grounding zero-shot objects.
>
> In the revised manuscript, we have integrated the above results and analysis (`highlighed in red; please see L940-L956 in Sec. G of the supplementary material`). Again, thanks!

---

> > ### Author Response · Authors · 2025-11-20
> > **Rebuttal (part 2/2)**
> >
> > > **Q4**: How much is each of ResNet and VidSwin feature helpful for the STVG performance?
> >
> > **Tab. D: Ablation of different features. "×" means that the model fails to converge.**
> > |   | Appearance Feature (ResNet) | Motion Feature (VidSwin)    | m_tIoU   | m_vIoU   |
> > |:-:|:------------------:|:--------------------:|:--------:|:--------:|
> > | ➊  | ✓                 |  | 35.29    | 8.88     |
> > | ➋ |  | ✓ | ×  | ×  |
> > | ➌ |     ✓             | ✓    | **35.83**| **9.47** |
> >
> > **A4**: Thanks for this insightful comment. As suggested, we show the experiments by studying the ResNet and VidSwin feature for performance in Tab. D. As shown in Tab. D, when using ResNet feature only (i.e., removing VidSwin feature; see ➊), the m_tIoU and m_vIoU are 35.29% and 8.88%, respectively, indicating that the ResNet-based appearance information is essential for grounding. This is expected because the ResNet-based appearance backbone is initialized from a pre-trained vision-language model, MDETR (Kamath et al.; ICCV 2021), and provides strong object-level semantics needed for target localization. In contrast, when using VidSwin feature only (i.e., removing ResNet feature; see ➋), the model fails to converge, and therefore no valid results can be reported. This indicates that, when using only VidSwin feature, the model loses its ability to ground textual references to visual regions and fails to converge meaningfully. When combining ResNet and VidSwin features, the model achieves the best performance with 35.83% m_tIoU and 9.47% m_vIoU, demonstrating that motion cues from VidSwin complement appearance information from ResNet and further enhance target localization.
> >
> > To clarify this point, we have added the above analysis in the revised version (`highlight in red; please see L957-L978 in Sec. G of the supplementary material`). Thanks again!
> >
> >
> >
> >
> > > **Q5**: A dataset has been collected hence any potential bias/responsible research practice could require more attention.
> >
> >
> > **A5**: Thanks for this careful comment. BOSTVG is constructed using videos sourced from YouTube under the ***Creative Commons license*** for ***research purpose only***. All videos and annotations are manually reviewed to avoid offensive or sensitive content, and we ensure diversity in both scenes and object categories during the data curation process.
> >
> >
> > As suggested, we have expanded the ethical considerations section in the revised manuscript by clearly describing the data collection protocol, potential limitations, and the intended research use of BOSTVG (`highlighted in red; please see L1125-L1133 in Sec. I of the supplementary material`). We appreciate the reviewer’s attention to responsible dataset creation. Again, thanks!

---

> > > ### Comment · Reviewer_81gP · 2025-11-23
> > > **Answer to rebuttal**
> > >
> > > I thank the authors for their answers and confirm they resolve my concerns. Therefore I am voting for accepting this paper.

---

> > > > ### Author Response · Authors · 2025-11-23
> > > >
> > > > Thank you for recognizing our work. Your constructive feedback and positive comments have been invaluable to us. Once again, thank you for your support and encouragement.

---

### Official Review · Reviewer_CK93 · 2025-10-22

**Soundness:** 3
**Presentation:** 3
**Contribution:** 3
**Rating:** 6
**Confidence:** 3

**Summary:**

Existing Spatio-Temporal Video Grounding (STVG) tasks are impractical, as they are conventionally trained to localize only a single target, even when multiple targets are referenced in the text.

This paper proposes a more practical new task, termed OmniSTVG (Omni-target Spatio-Temporal Video Grounding), which aims to localize all objects mentioned in the textual query. The primary contributions include:

New Task (OmniSTVG): Defining the aforementioned new direction of "omni-target" localization.

New Dataset (BOSTVG): To support this task, a large-scale, high-quality dataset comprising 10,000 videos was constructed, which has undergone multiple rounds of manual refinement.

New Baseline (OmniTube): An effective Transformer-based model is provided, which utilizes "text-guided queries" to concurrently localize all targets.

**Strengths:**

Pioneering a New Direction: The work identifies the critical limitation of existing STVG tasks (i.e., single-target grounding) and defines a more complex and practical new research direction (i.e., multi-target grounding).

Contribution of a Core Resource (BOSTVG): It provides the field's first large-scale (10,000 videos), high-quality benchmark dataset specifically dedicated to "omni-target" localization, establishing a core asset for advancing subsequent research.

Provision of a Strong Baseline (OmniTube): The study does not merely pose the problem but also delivers a well-designed (e.g., "text-guided queries") and empirically validated solution, offering a solid baseline for future work.

Rigorous and Solid Experimentation: The effectiveness of the model's constituent components is validated through exhaustive ablation studies, while comparative experiments underscore the uniqueness and necessity of the new BOSTVG dataset.

The manuscript is clearly and normatively written; the figures are highly comprehensible and meticulously detail the proposed methodology.

**Weaknesses:**

**Insufficient Analysis**:Regarding the benchmark, an analysis of the task's inherent difficulties is absent. The inclusion of illustrative examples to demonstrate these challenges would be beneficial.Furthermore, a deeper analysis is required as to why existing algorithms, including those compared in this study, cannot be directly or effectively applied to this benchmark.In the ablation study, the paper merely enumerates the functional contributions of individual algorithmic components without providing further in-depth analysis.

**Insufficient Representativeness of "Omni" (Data Distribution Imbalance)**:In Section 3.4 (Dataset Splits), the test set is partitioned into three groups: BOSTVG-Low (1-3 targets, 1566 samples), BOSTVG-Medium (4-6 targets, 273 samples), and BOSTVG-High (>7 targets, only 73 samples).The core objective of the paper is to address the "omni-target" problem; however, the proportion of high-density, genuinely complex "omni-target" samples (73) within the test set is extremely low (approximately 3.8% of the test set).This implies that the model's capability in handling complex, multi-target scenarios has not been sufficiently validated. The reported overall average performance (e.g., $m\_vIoU$) is likely dominated by the simpler "Low" group.

**Questionable Model Generalizability and Data Compatibility (Poor Generalizability)**:In Section 5.2 (Table 8: Ablation on Training Data), when attempting to merge BOSTVG with the existing single-target dataset HCSTVG-v2 for training (Row 3), performance paradoxically decreases slightly compared to training on BOSTVG alone (Row 1).Typically, augmenting the training data, even with related datasets, is expected to enhance model robustness.This performance degradation (which the authors attribute to "data inconsistency") may suggest that the model (OmniTube) or the annotation style of the dataset (BOSTVG) is prone to overfitting, hindering its ability to generalize or maintain compatibility with other data sources.

**Questions:**

see weakness

---

> ### Author Response · Authors · 2025-11-20
> **Rebuttal (part 1/3)**
>
> We thank the reviewer for these careful and helpful comments. We provide our responses below to answer the reviewer's questions.
>
> > **Q1**: Insufficient Analysis: **(a)** Regarding the benchmark, an analysis of the task's inherent difficulties is absent. The inclusion of illustrative examples to demonstrate these challenges would be beneficial. **(b)** Furthermore, a deeper analysis is required as to why existing algorithms, including those compared in this study, cannot be directly or effectively applied to this benchmark. **(c)** In the ablation study, the paper merely enumerates the functional contributions of individual algorithmic components without providing further in-depth analysis.
>
> **A1**: Sorry for the confusion. To better answer the reviewer's concern, we break down the question into three smaller questions **(a)**, **(b)**, and **(c)** and answer each of them below.
>
>
> For **(a)**: Thanks for this helpful comment. For the proposed BOSTVG benchmark, several inherent difficulties that go beyond conventional single-object STVG are introduced, including ***(i) multi-object grounding***: In our benchmark and task, the query often refers to multiple objects. To localize all the mentioned objects in query, the grounding model needs to understand complex relationships of multiple objects for accurate localization, which is inherently challenging; ***(ii) dynamic and crowded scenes:*** Compared to conventional single-object STVG benchmark, our proposed benchmark brings in more frequent and severe appearance changes, _e.g._, motion blur, partial or full occlusions among objects, and close object-interactions in videos, thereby demanding precise spatio-temporal alignment under visual ambiguity, particularly in the context of multi-object grounding; and ***(iii) more complex textual query:*** In our benchmark, the textual query is more complicated by containing more entities, due to its aim at localizing multiple objects. For robust performance, the grounding model needs to be equipped with strong scene-level reasoning. All these factors make BOSTVG a more realistic and challenging testbed for comprehensive video grounding.
>
> As suggested, we have added visual examples to better illustrate these inherent difficulties on our benchmark (`highlighted in red; please see L979-L1017 in Sec. G of the supplementary material`).
>
>
> For **(b)**: Thanks for this insightful comment. Existing STVG methods are designed for single-object grounding. For example, the recent CG-STVG mines visual context of the object to improve localization accuracy, but it can mine only the visual context of a single object and cannot be directly extended to multi-object scenarios. Moreover, existing methods lack explicit modeling of interactions among multiple objects, making them unable to resolve ambiguities when several similar or interacting  objects appear together, which is common in BOSTVG. In contrast, the proposed OmniTube is built from the ground up for omni-object grounding, supporting joint parsing, localization, and disambiguation of multiple targets. We have added this analysis in the revised manuscript (`highlighted in red; please see L1020-L1028 in Sec. G of the supplementary material`).
>
> For **(c)**: Thanks for this constructive comment. As suggested, we provide more analysis for each of our ablation studies below.
>
> ***(i) Ablation of spatial decoder***: To study different modules in spatial decoder, we conduct an ablation in Tab. 4. In our spatial decoder, the text-guided spatial query generation (**TG-SQG**) aims to learn target-specific spatial query for better interaction with multimodal feature. The spatial attention block (**SAB**) and temporal attention block (**TAB**) are applied to model spatial and temporal context within the video. As in Tab. 4, without TG-SQG, SAB and TAB, the m\_tIoU and m\_vIoU scores are 34.33\% and 8.25\% (➊). When SAB and TAB are added, the model better captures crucial spatial and temporal cues across frames, achieving comparable m\_tIoU of 34.13\% but higher m\_vIoU of 9.00\% (➊ _v.s._ ➋). This suggests that the spatial and temporal models in videos are crucial for target localization. More importantly, as shown in Tab. 4, when incorporating TG-SQG with either SAB or TAB, both m\_tIoU and m\_vIoU are improved (➊ _v.s._ ➌ and ➊ _v.s._ ➍). This indicates that the text-guided spatial query contains more discriminative target-specific information and is able to better interact with and learn from the multimodal features for target localization. Finally, when all modules are applied, we obtain the best performance with with 35.83\% m\_tIoU and 9.47\% m\_vIoU scores (➎), which validates the effectiveness of our decoder design in modeling spatial and temporal cues and learning target-specific query feature for improving grounding performance.

---

> ### Author Response · Authors · 2025-11-20
> **Rebuttal (part 2/3)**
>
> ***(ii) Ablation of temporal decoder***: To analyze temporal decoder, we conduct an ablation in Tab. 5. In our temporal decoder, the text-guided temporal query generation (**TG-TQG**) is designed to generate target-aware temporal query for localization, and the temporal attention block (**TAB**) is adopted to capture temporal relation across frames. As shown in Tab. 5, without using TG-TQG and TAB, the m\_tIoU and m\_vIoU scores are 26.06\% and 6.66\% (➊). When adding TAB for temporal modeling, we can see that the m\_tIoU and m\_vIoU scores are significantly improved to 35.00\% and 8.98\% (➊ _v.s._ ➋). This indicates that modeling temporal dependencies with TAB is essential for accurate temporal localization. When using TG-TQG alone, we achieve the similar m\_tIoU score of 26.00\% and m\_vIoU score of 6.82\% (➊ _v.s._ ➌). This suggests that although TG-TQG provides more discriminative target-aware cues, its benefit cannot be fully realized without temporal modeling. When combining TG-TQG and TAB, we obtain the best 35.83\% m\_tIoU and 9.47\% m\_vIoU scores (➍). This demonstrates that TG-TQG and TAB play complementary roles and their synergy leads to more accurate spatio-temporal grounding.
>
> ***(iii) Ablation of  different class predictions***: In OmniTube, rather than directly predicting the bounding‐box class for tube generation, we predict the position index of the class in the text as the bounding box class. To compare these two strategies, we conduct an ablation in Tab. 6. As shown in Tab. 6, predicting the position index yields consistently better performance (➊ _v.s._ ➋). We argue the reason is that, directly selecting from 287 classes is a challenging high‐cardinality classification problem, predicting the position index simplifies category localization by linking each box to its corresponding noun phrase in the text, thereby leading to better performance.
>
> ***(iv) Ablation of parameter M in the decoder***: In the decoder, $M$ controls how many video features are selected as relevant to the textual query. When $M$ is too small, the selected video features are insufficient to capture target-specific information from videos, which may results in performance degeneration. Conversely, when $M$ is too large, the selected video features may contain noise, leading to suboptimal performance. To study the effect of $M$, we conduct an ablation in Tab. 7. As shown in Tab. 7, we can observe that the best result is obtained when $M$ is 5 (➋), indicating that five query-relevant video features are sufficient for optimal grounding.
>
> ***(v) Ablation of training data***: OmniTube is trained with our distinct multi-object training data $BOSTVG_{Tra}$. To examine whether incorporating existing datasets can further improve performance, we evaluate the impact of adding existing HCSTVG-v2 with single-object data for training. As shown in Tab. 8, training with our multi-object $BOSTVG_{Tra}$ alone achieves the best results (➊), indicating that multi-object supervision by $BOSTVG_{Tra}$ is essential for OmniTube. In contrast, training solely with HCSTVG-v2 leads to the lowest performance (➋), because it lacks multi-object interactions and provides a weaker supervisory signal for the spatial–temporal grounding of multiple targets. When combining $BOSTVG_{Tra}$ and HCSTVG-v2 for training, the model performance slightly decreases (➌). This degradation is likely due to inconsistent training distributions for different tasks. HCSTVG-v2 contains only one human class and lacks multi-object queries. When merging $BOSTVG_{Tra}$ and HCSTVG-v2, the model is exposed to a large amount of samples with a very narrow semantic scope, which may bias learning toward human-centric patterns and weaken its ability to generalize across diverse object categories. Moreover, since HCSTVG-v2 lacks multi-object queries, it cannot provide supervision for joint localization of multiple entities. These two factors together result in the final slight performance drop, which also indicates that training on diverse, multi-class, multi-object data is essential for solving the omni-target problem.
>
> ***(vi) Impact of motion information***: To evaluate the effect of motion information, we conduct an ablation in Tab. 9. As shown in Tab. 9, incorporating motion feature consistently improves localization performance (➊ _v.s._ ➋). This is because motion encodes strong temporal signals that help link entities across frames and clarify the interactions among targets. Without motion cues, the model depends solely on appearance, which makes it harder to capture dynamic relationships and maintain temporal consistency. The results in Tab. 9 confirm that motion features play a complementary role to appearance cues and are essential for robust spatio-temporal grounding.
>
> We have integrated the above analysis of ablation studies in revised manuscript (`highlighted in red; see Sec. 5.2 for revised ablation studies`). Again, we thank the reviewer for these helpful comments.

---

> ### Author Response · Authors · 2025-11-20
> **Rebuttal (part 3/3)**
>
> > **Q2**: Insufficient Representativeness of "Omni" (Data Distribution Imbalance): In Section 3.4 (Dataset Splits), the test set is partitioned into three groups: BOSTVG-Low (1-3 targets, 1566 samples), BOSTVG-Medium (4-6 targets, 273 samples), and BOSTVG-High (≥7 targets, only 73 samples).The core objective of the paper is to address the "omni-target" problem; however, the proportion of high-density, genuinely complex "omni-target" samples (73) within the test set is extremely low (approximately 3.8% of the test set). This implies that the model's capability in handling complex, multi-target scenarios has not been sufficiently validated. The reported overall average performance (_e.g._, m_vIoU) is likely dominated by the simpler "Low" group.
>
> **A2**: We sincerely thank the reviewer for this insightful observation. Similar to other well-known datasets such as ImageNet (Deng et al.; CVPR 2009) and COCO (Lin et al.; ECCV 2014), the data distribution of our BOSTVG follows the long-tailed distribution, reflecting the natural frequency of multi-object scenarios in the real world. As a consequence, queries referring to 1–3 objects are more common, while those with ≥7 targets are rare but still included. To ensure fair evaluation across complexity levels, in addition to the overall average, we report separate metrics for BOSTVG-Low, -Medium, and -High (please see Tab. 3 in the paper).
>
> However, we completely understand reviewer's concern that the limited size of BOSTVG-High may restrict statistical confidence in extreme cases, and agree that adding more samples in BOSTVG-High is crucial for evaluating the model's capability in handling complex, multi-target scenarios. Addressing this, we will include more high-density examples (_e.g._, 200 more) in BOSTVG-High to enable more robust evaluation of scalability. Since this is beyond our current goal, we leave this to our future work.
>
>
> To make this point clearer, we have added the above clarification in the revised version (`highlighted in red; please see L1030-L1043 in Sec. G of the supplementary material`). Thanks again!
>
>
> > **Q3**: Questionable Model Generalizability and Data Compatibility (Poor Generalizability):In Section 5.2 (Table 8: Ablation on Training Data), when attempting to merge BOSTVG with the existing single-target dataset HCSTVG-v2 for training (Row 3), performance paradoxically decreases slightly compared to training on BOSTVG alone (Row 1). Typically, augmenting the training data, even with related datasets, is expected to enhance model robustness. This performance degradation (which the authors attribute to "data inconsistency") may suggest that the model (OmniTube) or the annotation style of the dataset (BOSTVG) is prone to overfitting, hindering its ability to generalize or maintain compatibility with other data sources.
>
>
> **Tab. B: Ablation of training data.**
> |   | **Training Data**                         | **# Objects** | **# Classes** | **Size** | **m_tIoU** | **m_vIoU** |
> |:-:|-------------------------------------------|:------------:|:------------:|:--------:|:----------:|:----------:|
> | ➊ | $BOSTVG_{Tra}$  |  1-10    |     287      | 8K    | 35.83  | 9.47  |
> | ➋| $HCSTVG$-$v2$ (training set)          |    1   |      1       | 10K    | 22.24      | 4.76       |
> | ➌| $BOSTVG_{Tra}$ + $HCSTVG$-$v2$ (training set)          |   1-10     |     287       | 18K    | 35.47      | 9.31       |
>
>
> **A3**: Sorry for the confusion. In our experiment (we copy the results in Tab. B here), when combining BOSTVG and HCSTVG-v2 for training, the performance slightly drops (➌ vs. ➊). We argue that the reason lies in the inconsistent training data for different tasks. As shown in Tab. B, HCSTVG-v2 contains only one class (_i.e._, Human) and focuses on single-object STVG. In contrast, our BOSTVG involves 287 object categories and requires modeling multiple interacting targets per query. When merging (the training subsets of) these datasets, the model is exposed to a large amount of samples with a very **narrow semantic scope**, which may bias learning toward human-centric patterns and weaken its ability to generalize across diverse object categories. Moreover, since HCSTVG-v2 lacks multi-object queries, it cannot provide supervision for joint localization of multiple entities. These two factors together result in the final slight performance drop, which also indicates that **training on diverse, multi-class, multi-object data is essential** for solving the omni-target problem. We believe that this result underscores the importance of our dataset design: BOSTVG is not just an extension of existing work, but a necessary shift toward richer, more complex training signals.
>
> we have inlcuded the above clarification and anlaysis in the revised version (`highlighted in red; please see L481-L503 in Sec. 5.2`). Again, Thanks!

---

### Official Review · Reviewer_nfHN · 2025-10-31

**Soundness:** 3
**Presentation:** 4
**Contribution:** 3
**Rating:** 8
**Confidence:** 3

**Summary:**

The paper proposes a new video grounding task, OmniSTVG, which differs from the traditional STVG task in that it can localize all targets mentioned in the textual query as well as the interactive relationships existing among these targets. To this end, the paper constructs a large-scale dataset BOSTVG and presents a simple yet effective model named OmniTube.

**Strengths:**

- The task proposed in the paper expands the scope of the traditional STVG task. Moreover, the proposed dataset has a wide range of sources and is built using a relatively rigorous manual annotation method, combining both scale and quality.
- The paper puts forward a simple and effective baseline, and has implemented and compared several public models.
- The paper provides detailed descriptions of details, making it easy to follow.

**Weaknesses:**

The paper does not mention the performance of multimodal large language models (MLLMs) on this task.

**Questions:**

Why is the performance of any multimodal large language models on this dataset not provided?

---

> ### Author Response · Authors · 2025-11-20
> **Rebuttal**
>
> We thank the reviewer for helpful comments on our work and provide our responses below to the reviewer's questions.
>
> >  **Q1**: The paper does not mention the performance of multimodal large language models (MLLMs) on this task. Why is the performance of any multimodal large language models on this dataset not provided?
>
> **Tab. A: Comparison with existing mllm-based methods on BOSTVG$_\textbf{Tst}$-Full.**
> || Method            | m\_tIoU | m\_vIoU | vIoU\@0.3 | vIoU\@0.5 |
> |---|-------------------|:---------:|:---------:|:----------:|:----------:|
> |➊| Gemini-2.5 Pro    |  35.5 | 7.2 |  5.1 |  0.5 |
> |➋| Qwen2.5-VL      |  15.2   | 1.6  |   0.0   |   0.0  |
> |➌| Qwen2.5-VL-SFT      | 29.1   | 5.2   | 2.2    | 0.1  |
> |➍| OmniTube (ours)    | 35.8 | 9.5 | 6.2 | 0.9 |
>
> **A1**: Thanks for this thoughtful comment. We agree that including MLLM-based method is crucial and valuable. As suggested, we have added experiments evaluating two popular MLLMs, including Gemini-2.5 Pro and Qwen2.5-VL, on OmniSTVG. Please note that, since Gemini-2.5 Pro is not open-source, we directly apply Gemini-2.5 Pro to predict spatial-temporal locations of targets in videos without any tuning. For Qwen2.5-VL, we report results of two versions, include zero-shot version (named Qwen2.5-VL) without tuning and supervised fine-tuned version (named Qwen2.5-VL-SFT). The results and comparison on BOSTVG$_\textbf{Tst}$-Full are shown in Tab. A. From Tab. A, we observe that, our proposed OmniTube (➍) outperforms all MLLM-based methods (➊: Gemini-2.5 Pro, ➋: Qwen2.5-VL without tuning, ➌: Qwen2.5-VL-SFT with supervised fine-tuning). We argue that the reasons why MLLM-based methods underperform are two fold: (1) without task-specific tuning (see ➊ and ➋), MLLMs-based method are not equipped to perform fine-grained spatio-temporal localization for multiple objects in videos; and (2) even with fine-tuning (see ➌), the training objective of MLLM treats structured outputs such as timestamps and bounding boxes as discrete text tokens, making them particularly sensitive to even small lexical differences (_e.g._, “1–9s” versus “0–8s”). These findings highlight the challenges of our OmniSTVG task and the necessity of dedicated architectures like the proposed OmniTube.
>
> We have incorporated the new results and corresponding analysis in the revised manuscript (`highlighted in red; please see L918-L938 in Sec. G of the supplementary material`). Again, thanks for this helpful comment!

---

> > ### Comment · Reviewer_nfHN · 2025-11-25
> >
> > I have no further questions and will continue to hold my positive ratings.

---

> > > ### Author Response · Authors · 2025-11-25
> > >
> > > Thank you very much for your positive comment and for taking the time to review our work. We truly appreciate your feedback.

---

### Meta-Review · Program_Chairs · 2026-01-05

**Summary:**

This paper introduces OmniSTVG, a new formulation of spatio-temporal video grounding that requires grounding all objects mentioned in a free-form query, along with a large-scale benchmark (BOSTVG) and a corresponding baseline method (OmniTube). Reviewers generally found the problem formulation timely and the dataset contribution valuable, with comprehensive experiments and competitive performance.

The main concerns raised across reviews relate to (i) the representativeness of the most challenging multi-target test subsets, (ii) certain modeling assumptions such as shared temporal segments across objects, and (iii) clarity in positioning the baseline relative to prior STVG pipelines. After considering the reviews and the rebuttal, these issues are judged to be non-blocking and do not outweigh the overall contribution of the paper.

**This paper is conditionally accepted provided the authors do the following for camera ready**:
[Ethics concerns] In addition to the details of data collection, please detail any IRB approvals/exemptions needed for annotator to remove sensitive or offensive content.

**Reviewer Concerns:**

Concerns addressed by the rebuttal:
The rebuttal meaningfully addressed several key reviewer concerns. In particular, the authors added comparisons to strong MLLM-based baselines (e.g., Gemini and Qwen-VL), which contextualize the proposed method against recent large multimodal models and strengthen the empirical evaluation.
The rebuttal also clarified the construction of the BOSTVG benchmark, including the long-tail distribution of multi-object queries and the rationale behind the “High” subset, and provided additional discussion on dataset compatibility issues observed when mixing BOSTVG with prior benchmarks.

Existing concerns:
Some concerns remain regarding the strength of certain modeling assumptions, notably the assumption that all queried objects share the same temporal segment, which may limit applicability to more general multi-object narratives.
In addition, the size of the most challenging multi-target subset is relatively small, and clearer attribution of what is novel versus adopted in the baseline design would further improve the presentation.
However, these limitations primarily affect generality and clarity rather than the correctness or significance of the contribution, and they do not undermine the validity of the proposed benchmark or the relevance of the task.

**Reviewer Scores:**

Reviewer nfHN: Likely to maintain their original score, as their main requests for stronger baselines and clearer evaluation have been addressed in the rebuttal.

Reviewer CK93: Likely to maintain their score. While they noted dataset and modeling assumptions, they viewed these as limitations rather than fundamental flaws.

Reviewer 81gP: Might slightly increase or maintain their score after discussion, given the added MLLM comparisons and clarifications regarding benchmark design, though some reservations about temporal assumptions may remain.

---

### Decision · Program_Chairs · 2026-01-26

**Decision:**

Accept (Poster)

**Comment:**

Conditions for acceptance have been satisfied.